# Out-of-Distribution Generalization in Kernel Regression

**Abdulkadir Canatar**
Department of Physics
Harvard University
Cambridge, MA 02138
`canatara@g.harvard.edu`

**Blake Bordelon**
John A. Paulson School of Engineering and Applied Sciences
Harvard University
Cambridge, MA 02138
`blake_bordelon@g.harvard.edu`

**Cengiz Pehlevan**
John A. Paulson School of Engineering and Applied Sciences
Harvard University
Cambridge, MA 02138
`cpehlevan@g.harvard.edu`

## Abstract

In real word applications, the data generating process for training a machine learning model often differs from what the model encounters in the test stage. Understanding how and whether machine learning models generalize under such distributional shifts remains a theoretical challenge. Here, we study generalization in kernel regression when the training and test distributions are different using the replica method from statistical physics. We derive an analytical formula for the out-of-distribution generalization error applicable to any kernel and real datasets. We identify an overlap matrix that quantifies the mismatch between distributions for a given kernel as a key determinant of generalization performance under distribution shift. Using our analytical expressions we elucidate various generalization phenomena including possible improvement in generalization when there is a mismatch. We develop procedures for optimizing training and test distributions for a given data budget to find best and worst case generalizations under the shift. We present applications of our theory to real and synthetic datasets and for many kernels. We compare results of our theory applied to Neural Tangent Kernel with simulations of wide networks and show agreement. We analyze linear regression in further depth.

## 1   Introduction

Machine learning models are trained to accurately predict on previously unseen samples. A central goal of machine learning theory has been to understand this generalization performance [1, 2]. While most of the theory in this domain focused on generalizing in-distribution, i.e. when the training examples are sampled from the same distribution as that of the test examples, in real world applications there is often a mismatch between the training and the test distributions [3]. Such difference, even when small, may lead to large effects in generalization performance [4, 5, 6, 7, 8].

In this paper, we provide a theory of out-of-distribution (OOD) generalization for kernel regression [9, 10, 11] applicable to any kernel and real datasets. Besides being a popular machine learning method itself, kernel regression is recovered from an infinite width limit of neural networks [12], making our results relevant to understanding OOD generalization in deep neural networks. Indeed, it

35th Conference on Neural Information Processing Systems (NeurIPS 2021).

has been argued that understanding generalization in kernel methods is necessary for understanding generalization in deep learning [13].

Using methods from statistical physics [14] and building on techniques developed in a recent line of work [15, 16], we obtain an analytical expression for generalization in kernel regression when test data is drawn from a different distribution than the training distribution. Our results apply to average or typical case learning curves and describe numerical experiments well including those on real data. In contrast, most previous works focus on worst-case OOD generalization bounds in the spirit of statistical learning theory [17, 18, 19, 20, 21, 22, 23].

We show examples of how mismatched training and test distributions affect generalization and in particular, demonstrate that the mismatch may improve test performance. We use our analytical expression to develop a procedure for minimizing generalization error with respect to the training/test distributions and present applications of it. We study OOD generalization with a linear kernel in detail and present examples of how dimensional reduction of the training distribution can be helpful in generalization, including shifting of a double-descent peak [24]. We present further analyses of various models in Supplementary Information (SI).

## 2 OOD Generalization Error for Kernel Regression from the Replica Method

We consider kernel regression in the setting where training examples are sampled i.i.d. from a distribution $p(\mathbf{x})$ but the test examples are drawn from a different distribution $\tilde{p}(\mathbf{x})$. We derive our main analytical formula for the generalization error here, and illustrate and discuss its implications in the following sections.

### 2.1 Problem setup

We consider a training set $\mathcal{D} = \{(\mathbf{x}^\mu, y^\mu)\}_{\mu=1}^P$ of size $P$ where the $D$-dimensional inputs $\mathbf{x} \in \mathbb{R}^D$ are drawn independently from a training distribution $p(\mathbf{x})$ and the noisy labels are generated from a target function $y^\mu = \bar{f}(\mathbf{x}^\mu) + \epsilon^\mu$ where the noise covariance is $\mathbb{E}[\epsilon^\mu \epsilon^\nu] = \varepsilon^2 \delta_{\mu\nu}$. Kernel regression model is trained through minimizing the training error:

$$f_{\mathcal{D}}^* = \underset{f \in \mathcal{H}}{\arg\min} \frac{1}{2} \sum_{\mu=1}^P \left( f(\mathbf{x}^\mu) - y^\mu \right)^2 + \lambda \langle f, f \rangle_{\mathcal{H}}, \tag{1}$$

where subscript $\mathcal{D}$ emphasizes the dataset dependence. Here $\mathcal{H}$ is a Reproducing Kernel Hilbert Space (RKHS) associated with a positive semi-definite kernel $K(\mathbf{x}, \mathbf{x}') : \mathbb{R}^D \times \mathbb{R}^D \to \mathbb{R}$, and $\langle ., . \rangle_{\mathcal{H}}$ is the Hilbert inner product.

The generalization error on the test distribution $\tilde{p}(\mathbf{x})$ is given by $E_g(\mathcal{D}) = \int d\mathbf{x}\, \tilde{p}(\mathbf{x}) \left( f_{\mathcal{D}}^*(\mathbf{x}) - \bar{f}(\mathbf{x}) \right)^2$. We note that this quantity is a random variable whose value depends on the sampled training dataset. We calculate its average over the distribution of all datasets with sample size $P$:

$$E_g = \mathbb{E}_{\mathcal{D}} \left[ \int d\mathbf{x}\, \tilde{p}(\mathbf{x}) \left( f_{\mathcal{D}}^*(\mathbf{x}) - \bar{f}(\mathbf{x}) \right)^2 \right]. \tag{2}$$

As $P$ grows, we expect fluctuations around this average to fall and $E_g(\mathcal{D})$ to concentrate around $E_g$. We will demonstrate this concentration in simulations.

### 2.2 Overview of the calculation

We calculate $E_g$ using the replica method from statistical physics of disordered systems [14, 25]. Details of calculations are presented in Section SI.1. Here we give a short overview. Readers may choose to skip this part and proceed to the main result in Section 2.3.

Our goal is to calculate the dataset averaged estimator $f^*(\mathbf{x}) \equiv \mathbb{E}_{\mathcal{D}} f_{\mathcal{D}}^*(\mathbf{x})$ and its covariance $\mathbb{E}_{\mathcal{D}} \left[ \left( f_{\mathcal{D}}^*(\mathbf{x}) - f^*(\mathbf{x}) \right) \left( f_{\mathcal{D}}^*(\mathbf{x}') - f^*(\mathbf{x}') \right) \right]$. From these quantities, we can reconstruct $E_g$ using

the bias-variance decomposition of $E_g = B + V$, where $B = \int d\mathbf{x}\, \tilde{p}(\mathbf{x}) \left( f^*(\mathbf{x}) - \bar{f}(\mathbf{x}) \right)^2$ and $V = \mathbb{E}_{\mathcal{D}} \left[ \int d\mathbf{x}\, \tilde{p}(\mathbf{x}) \left( f_{\mathcal{D}}^*(\mathbf{x}) - f^*(\mathbf{x}) \right)^2 \right]$.

For this purpose, it will be convenient to work with a basis in the RKHS defined by Mercer's theorem [26]. One can find a (possibly infinite dimensional) complete orthonormal basis $\{\phi_\rho\}_{\rho=1}^M$ for $L^2(\mathbb{R}^D)$ with respect to the training probability distribution $p(\mathbf{x})$ due to Mercer's theorem [26] such that

$$K(\mathbf{x}, \mathbf{x}') = \mathbf{\Phi}(\mathbf{x})^\top \mathbf{\Lambda} \mathbf{\Phi}(\mathbf{x}'), \quad \int d\mathbf{x}\, p(\mathbf{x}) \mathbf{\Phi}(\mathbf{x}) \mathbf{\Phi}(\mathbf{x})^\top = \mathbf{I}, \tag{3}$$

where we defined $M \times M$ diagonal eigenvalue matrix $\mathbf{\Lambda}_{\rho\gamma} = \eta_\rho \delta_{\rho\gamma}$ and the column vector $\mathbf{\Phi}(\mathbf{x}) = \left( \phi_1(\mathbf{x}), \phi_2(\mathbf{x}), \ldots, \phi_M(\mathbf{x}) \right)$. Eigenvalues and eigenfunctions satisfy the integral eigenvalue equation

$$\int d\mathbf{x}'\, p(\mathbf{x}') K(\mathbf{x}, \mathbf{x}') \phi_\rho(\mathbf{x}') = \eta_\rho \phi_\rho(\mathbf{x}). \tag{4}$$

We note that the kernel might not express all eigenfunctions if the corresponding eigenvalues vanish. Here we assume that all kernel eigenvalues are non-zero for presentation purposes, however in Section SI.1 we consider the full case. We also define a feature map via the column vector $\mathbf{\Psi}_\rho(\mathbf{x}) \equiv (\mathbf{\Lambda}^{1/2} \mathbf{\Phi}(\mathbf{x}))_\rho = \sqrt{\eta_\rho} \phi_\rho(\mathbf{x})$ so that $\langle \mathbf{\Psi}(\mathbf{x}), \mathbf{\Psi}(\mathbf{x}) \rangle_{\mathcal{H}} = \mathbf{I}$. The complete basis $\mathbf{\Phi}(\mathbf{x})$ can be used to decompose the target function:

$$\bar{f}(\mathbf{x}) = \bar{\mathbf{a}}^\top \mathbf{\Phi}(\mathbf{x}) = \bar{\mathbf{w}}^\top \mathbf{\Psi}(\mathbf{x}), \tag{5}$$

where $\bar{w} = (\mathbf{\Lambda}^{-1/2} \bar{\mathbf{a}})$, and $\bar{\mathbf{a}}$ and $\bar{\mathbf{w}}$ are vectors of coefficients. A function belongs to the RKHS $\mathcal{H}$ if it has finite Hilbert norm $\langle f, f \rangle_{\mathcal{H}}^2 = \mathbf{a}^\top \mathbf{\Lambda}^{-1} \mathbf{a} < \infty$.

With this setup, denoting the estimator as $f(\mathbf{x}) = \mathbf{w}^\top \mathbf{\Psi}(\mathbf{x})$ and the target function $\bar{f}(\mathbf{x})$ as given in Eq.(5), kernel regression problem reduces to minimization of the energy function $H(\mathbf{w}; \mathcal{D}) \equiv \frac{1}{2\lambda} \sum_{\mu=1}^P \left( (\bar{\mathbf{w}} - \mathbf{w})^\top \mathbf{\Psi}(\mathbf{x}^\mu) + \epsilon^\mu \right)^2 + \frac{1}{2} \|\mathbf{w}\|_2^2$. with optimal estimator weights $\mathbf{w}_{\mathcal{D}}^* = \arg\min_{\mathbf{w}} H(\mathbf{w}; \mathcal{D})$. We again emphasize the dependence of the optimal estimator weights $\mathbf{w}_{\mathcal{D}}^*$ to the particular choice of training data $\mathcal{D}$.

We map this problem to statistical mechanics by defining a Gibbs distribution $\propto e^{-\beta H(\mathbf{w}; \mathcal{D})}$ over estimator weights $\mathbf{w}$ which concentrates around the kernel regression solution $\mathbf{w}_{\mathcal{D}}^*$ as $\beta \to \infty$. This can be used to calculate any function $O(\mathbf{w}^*; \mathcal{D})$ by the following trick:

$$O(\mathbf{w}^*; \mathcal{D}) = \lim_{\beta \to \infty} \frac{\partial}{\partial J} \log Z[J; \beta, \mathcal{D}] \big|_{J=0}, \quad Z[J; \beta, \mathcal{D}] = \int d\mathbf{w}\, e^{-\beta H(\mathbf{w}; \mathcal{D}) + JO(\mathbf{w})}, \tag{6}$$

where $Z$ is the normalizer of the Gibbs distribution, also known as the partition function. Next, we want to compute the average of $\mathbb{E}_{\mathcal{D}} O(\mathbf{w}^*; \mathcal{D})$ which requires computing $\mathbb{E}_{\mathcal{D}} \log Z$. Further, experience from the study of physics of disordered systems suggests that the logarithm of the partition function concentrates around its mean (is self-averaging) for large $P$ [14], making our theory applicable to the typical case. However, this average is analytically hard to calculate due to the partition function appearing inside the logarithm. To proceed, we resort to the replica method from statistical physics [14], which uses the equality $\mathbb{E}_{\mathcal{D}} \log Z = \lim_{n \to 0} \frac{\mathbb{E}_{\mathcal{D}} Z^n - 1}{n}$. The method proceeds by calculating the right hand side for integer $n$, analytically continuing the resulting expression to real valued $n$, and performing the limit. While non-rigorous, the replica method has been successfully used in the study of the physics of disordered systems [14] and machine learning theory [27]. A crucial step in our computation is approximating $\mathbf{w}^\top \mathbf{\Psi}(\mathbf{x})$ as a Gaussian random variable via its first and second moments when averaged over the training distribution $p(\mathbf{x})$. It has been shown that this approximation yields perfect agreement with experiments [28, 15, 16]. Details of our calculation is given in Supplementary Section SI.1.

## 2.3 Main Result

The outcome of our statistical mechanics calculation (Section SI.1) which constitutes our main theoretical result is presented in the following proposition.

**Proposition 1.** *Consider the kernel regression problem outlined in Section 2.1, where the model is trained on $p(\mathbf{x})$ and tested on $\tilde{p}(\mathbf{x})$. Consider the Mercer decomposition of the RKHS kernel*

$K(\mathbf{x}, \mathbf{x}') = \mathbf{\Phi}(\mathbf{x})^\top \mathbf{\Lambda} \mathbf{\Phi}(\mathbf{x}')$, *where we defined* $M \times M$ *(M possibly infinite) diagonal eigen-value matrix* $\mathbf{\Lambda}_{\rho\gamma} = \eta_\rho \delta_{\rho\gamma}$ *and the column vector* $\mathbf{\Phi}(\mathbf{x}) = \big(\phi_1(\mathbf{x}), \phi_2(\mathbf{x}), \ldots, \phi_M(\mathbf{x})\big)$, *with* $\int d\mathbf{x}\, p(\mathbf{x}) \mathbf{\Phi}(\mathbf{x}) \mathbf{\Phi}(\mathbf{x})^\top = \mathbf{I}$. *Also consider an expansion of the target function in the Mercer basis* $\bar{f}(\mathbf{x}) = \bar{\mathbf{a}}^\top \mathbf{\Phi}(\mathbf{x})$.

*The dataset averaged out-of-distribution generalization error is given by:*

$$E_g = E_g^{0,p(\mathbf{x})} + \frac{\gamma' - \gamma}{1 - \gamma}\varepsilon^2 + \kappa^2 \bar{\mathbf{a}}^\top (P\mathbf{\Lambda} + \kappa\mathbf{I})^{-1} \mathscr{O}'(P\mathbf{\Lambda} + \kappa\mathbf{I})^{-1}\bar{\mathbf{a}},$$

$$\kappa = \lambda + \kappa \operatorname{Tr}(P + \kappa\mathbf{\Lambda}^{-1})^{-1}, \quad \gamma = P\operatorname{Tr}(P + \kappa\mathbf{\Lambda}^{-1})^{-2}, \quad \gamma' = P\operatorname{Tr}\mathscr{O}(P + \kappa\mathbf{\Lambda}^{-1})^{-2}, \quad (7)$$

*where* $\kappa$ *must be solved self-consistently, and we defined the* $M \times M$ *overlap matrix*

$$\mathscr{O}_{\rho\gamma} = \int d\mathbf{x}\, \tilde{p}(\mathbf{x})\phi_\rho(\mathbf{x})\phi_\gamma(\mathbf{x}), \quad \mathscr{O}' = \mathscr{O} - \frac{1 - \gamma'}{1 - \gamma}\mathbf{I}. \tag{8}$$

*Here* $E_g^{0,p(\mathbf{x})}$ *denotes the generalization error when both training and test distributions are matched to* $p(\mathbf{x})$ *and is given by:*

$$E_g^{0,p(\mathbf{x})} = \frac{\gamma}{1 - \gamma}\varepsilon^2 + \frac{\kappa^2}{1 - \gamma}\bar{\mathbf{a}}^\top (P\mathbf{\Lambda} + \kappa\mathbf{I})^{-2}\bar{\mathbf{a}}, \tag{9}$$

*which coincides with the findings of [15, 16]. Further, the expected estimator is:*

$$f^*(\mathbf{x}; P) = \sum_\rho \frac{P\eta_\rho}{P\eta_\rho + \kappa}\bar{a}_\rho \phi_\rho(\mathbf{x}). \tag{10}$$

Several remarks are in order.

*Remark* 1. Our result is general in that it applies to any kernel, data distribution and target function. When applied to kernels arising from the infinite width limit of neural networks [12], the information about the architecture of the network is in the kernel spectrum $\mathbf{\Lambda}$ and the target weights $\bar{\mathbf{a}}$ obtained by projecting the target function onto kernel eigenbasis.

*Remark* 2. Formally, the replica computation requires a thermodynamic limit in which $P \to \infty$, where variations in $E_g$ due to the sampling of the training set become negligible. The precise nature of the limit depends on the kernel and the data distribution, and includes scaling of other variables such as $D$ and $M$ with $P$. We will give examples of such limits in SI. However, we observe in simulations that our formula predicts average learning curves accurately for even as low as a few samples.

*Remark* 3. We recover the result obtained in [15, 16] when the training and test distributions are the same ($\mathscr{O} = \mathbf{I}$ which implies $E_g = E_g^{0,p(\mathbf{x})}$).

*Remark* 4. *Mismatched training and test distributions may improve test error.* Central to our analysis is the shifted overlap matrix $\mathscr{O}'$ which may have negative eigenvalues and hence cause better generalization performance when compared to in-distribution generalization.

*Remark* 5. The estimator $f^*$ only depends on the training distribution and our theory predicts that the estimator eventually approaches to the target $f^* \to \bar{f}$ for large enough $P$ which performs perfectly on the training distribution unless there are out-of-RKHS components in target function. The latter case is studied in depth in Section SI.1.

*Remark* 6. While stated for a scalar output, our result can be trivially generalized to a vector output by simply adding the error due to each component of the output vector, as described in [15].

Next we analyze this result by studying various examples.

## 3 Applications to Real Datasets

First, we test our theory on real datasets and demonstrate its broad applicability. To do that, we define Dirac training and test probability measures for a fixed dataset $\mathcal{D}$ of size $M$ with some point mass on each of the data points $p(\mathbf{x}) = \sum_{\mathbf{x}^\mu \in \mathcal{D}} p^\mu \delta(\mathbf{x} - \mathbf{x}^\mu)$ and $\tilde{p}(\mathbf{x}) = \sum_{\mathbf{x}^\mu \in \mathcal{D}} \tilde{p}^\mu \delta(\mathbf{x} - \mathbf{x}^\mu)$. With this measure, the kernel eigenvalue problem in Eq.(4) becomes $\mathbf{K}\operatorname{diag}(\boldsymbol{p})\mathbf{\Phi} = \mathbf{\Phi}\mathbf{\Lambda}$, where $\operatorname{diag}(\boldsymbol{p})$ denotes the diagonal matrix of probability masses of each data point. In this setup the number of

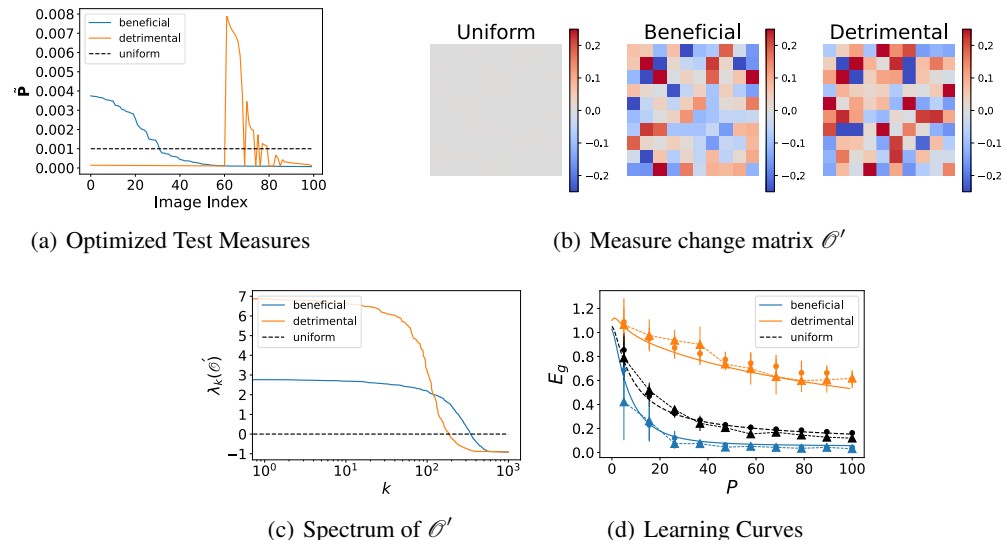

(a) Optimized Test Measures

(b) Measure change matrix $\mathscr{O}'$

(c) Spectrum of $\mathscr{O}'$

(d) Learning Curves

Figure 1: Shifts in the test distribution can be understood through the matrix $\mathscr{O}'$. (a) We ran gradient descent (beneficial) and gradient ascent (detrimental) on the theoretical generalization error $E_g$ with respect to the test measure on 1000 MNIST images. (a) The final probability of the first 100 images sorted by probability mass for the beneficial measure. (b) For these three distributions (uniform, beneficial, detrimental), the measure change matrices $\mathscr{O}'$ are plotted for the top 10 modes. The beneficial and detrimental matrices are roughly opposite. (c) The spectrum of $\mathscr{O}'$ reveals both negative and positive eigenvalues which demonstrates that increase or a decrease in the generalization error is possible. (d) The learning curves show the predicted generalization error (lines) along with trained neural network (triangles) and corresponding NTK regression experiments (dots). Error bars indicate standard deviation over 35 random trials.

eigenfunctions, or eigenvectors, are equal to the size of entire dataset $\mathcal{D}$. Once the eigenvalues $\mathbf{\Lambda}$ and the eigenvectors $\mathbf{\Phi}$ have been identified, we compute the target function coefficients by projecting the target data $\mathbf{y}_c$ onto these eigenvectors $\bar{\mathbf{a}}_{\mathbf{c}} = \mathbf{\Phi}^\top \text{diag}(\boldsymbol{p})\mathbf{y}_c$ for each target $c = 1, \ldots, C$. Once all of these ingredients are obtained, theoretical learning curves can be computed using Proposition 1. This procedure is similar to the one used in [15, 16].

## 3.1 Shift in test distribution may help or hurt generalization

To analyze the influence of distribution shift on generalization error, we first study the problem of a fixed training distribution and a shifting test distribution. While one may naively expect that a shift in the test distribution would always result in worse performance, we demonstrate that mismatch can be either beneficial or detrimental to generalization. A similar point was made in [4].

Since our generalization error formula is a differentiable function with respect to the training measure, we numerically optimize $E_g$ with respect to $\{\tilde{p}^\mu\}$ using gradient descent to find beneficial test measures where the error is smaller than for the case where training and test measures are matched. Similarly, we perform gradient ascent to find detrimental test measures, which give higher test error than the matched case.

The result of this procedure is shown in Figure 1. We perform gradient descent (labeled beneficial) and gradient ascent (labeled detrimental) on $M = 1000$ MNIST digits 8's and 9's, where target outputs are binary $\{-1, 1\}$ corresponding to two digits. We use a depth 3 ReLU fully-connected neural network with 2000 hidden units at each layer and its associated Neural Tangent Kernel (NTK) which are computed using the NeuralTangents API [29]. The probability mass assigned to the first 50 points are provided in Figure 1(a). We see that points given high mass for the beneficial test distribution are given low probability mass for the detrimental test distribution and vice versa. We plot the measure change matrix $\mathscr{O}'$ which quantifies the change in the generalization error due to distribution shift. Recall that in the absence of noise, $E_g$ is of the form $E_g = E_g^{\text{matched}} + \boldsymbol{v}^\top \mathscr{O}' \boldsymbol{v}$ for

**Algorithm 1:** Optimizing Training Measure at sample size $P$

---

**Result:** GET_LOSS($\boldsymbol{z} \in \mathbb{R}^M$, $\boldsymbol{K} \in \mathbb{R}^{M \times M}$, $\boldsymbol{y} \in \mathbb{R}^{M \times C}$, $\lambda \in \mathbb{R}_+$)

    Compute normalized distribution $\boldsymbol{p} = \text{softmax}(\boldsymbol{z})$;

    Diagonalize on Train Measure $\boldsymbol{K}\,\text{diag}(\boldsymbol{p})\boldsymbol{\Phi} = \boldsymbol{\Phi}\boldsymbol{\Lambda}$ with $\boldsymbol{\Phi}^\top \text{diag}(\boldsymbol{p})\boldsymbol{\Phi} = \boldsymbol{I}$ ;

    Get Target Function Weights $\overline{\mathbf{a}} = \boldsymbol{\Phi}^\top \text{diag}(\boldsymbol{p})\boldsymbol{y}$;

    Get Overlap Matrix $\mathscr{O} = \frac{1}{M}\boldsymbol{\Phi}^\top \boldsymbol{\Phi}$ ;

    Solve Implicit Equation $\kappa = \text{ODE-INT}\left[\dot{\kappa} = \lambda + \kappa \sum_k \frac{\Lambda_{kk}}{\Lambda_{kk}P + \kappa} - \kappa\right]$ ;

    **return** $E_g(\kappa, P, \boldsymbol{\Lambda}, \mathscr{O})$ (see Proposition 1);

;

**Result:** OPT_MEASURE($\boldsymbol{K} \in \mathbb{R}^{M \times M}$, $\boldsymbol{y} \in \mathbb{R}^M$, $P$, $T$ ,$\eta$, $\lambda$)

    Initialize $\boldsymbol{z} = \mathbf{0} \in \mathbb{R}^M$ , $t = 0$;

    Diagonalize Kernel on Uniform Measure $\frac{1}{M}\boldsymbol{K} = \tilde{\boldsymbol{\Phi}}\tilde{\boldsymbol{\Lambda}}\tilde{\boldsymbol{\Phi}}^\top$;

    **while** $t < T$ **do**

        $\boldsymbol{z} = \boldsymbol{z} - \eta\,\text{GRAD}[\text{GET\_LOSS}(\boldsymbol{z}, \boldsymbol{K}, \boldsymbol{y}, \lambda)]$;

        $t = t + 1$;

    **end**

    **return** $\text{softmax}(\boldsymbol{z})$ ;

---

a vector $\boldsymbol{v}$ which depends on the task, kernel, training measure and sample size $P$. Depending on the eigenvalues and eigenvectors of $\mathscr{O}'$, and the vector $\boldsymbol{v}$, the quadratic form $\boldsymbol{v}^\top \mathscr{O}' \boldsymbol{v}$ may be positive or negative, resulting in improvement or detriment in the generalization error. In Figure 1(b), we see that the $\mathscr{O}'$ matrix for the beneficial test measure appears roughly opposite that of the detrimental measure. This is intuitive since it would imply the change in the generalization error to be opposite in sign for beneficial and detrimental measures $\boldsymbol{v}^\top \left(-\mathscr{O}'\right)\boldsymbol{v} = -\boldsymbol{v}^\top \mathscr{O}'\boldsymbol{v}$. The spectrum of the matrix, shown in Figure 1(c), reveals that it possesses both positive and negative eigenvalues. We plot the theoretical and experimental learning curves in Figure 1(d). As promised, the beneficial (blue) measure has lower generalization error while the detrimental measure (orange) has higher generalization error than the matched case (black). Experiments show excellent match to our theory. In Section SI.2, we present another application of this procedure to adversarial attacks during testing.

## 3.2  Optimized Training Measure for Digit Classification with Wide Neural Networks

Often in real life tasks test distribution is fixed while one can alter the training distribution for more efficient learning. Therefore, we next study how different training distributions affect the generalization performance of kernel regression when the test distribution is fixed. We provide pseudocode in Algorithm 1 which describes a procedure to optimize the expected generalization error with respect to a training distribution of $P$ data points. This optimal training distribution has been named the dual distribution to $\tilde{p}(x)$ [4]. All of the operations in the computation of $E_g$ support automatic differentiation with respect to logits $\boldsymbol{z} \in \mathbb{R}^M$ that define the training distribution through $\text{softmax}(\boldsymbol{z})$, allowing us to perform gradient descent on the training measure [30].

As an example, we fix the test distribution to be uniform MNIST digits for 8's and 9's and optimize over the probability mass of each unique digit to minimize $E_g$, again using a depth 3 ReLU neural tangent kernel and binary targets $\{-1, 1\}$. Our results indicate that it is possible to reduce generalization when the training distribution is chosen differently than the uniform test distribution (See Section SI.2 for extended discussion). In Figure 2, we optimize $E_g$ for $P = 30$ training samples to extract the optimal training distribution on NTK. We observe that the high mass training digits are closer to the center of their class, Figure 2(e). However we also find that optimizing the distribution for different values of $P$ may cause worse generalization beyond the $P$ used to optimize the distribution Figure 2(g, h).

We also test our results on neural networks, exploiting a correspondence between kernel regression with the Neural Tangent Kernel, and training infinitely wide neural networks [12]. Figure 2(i) shows that our theory matches experiments with ReLU networks of modest width 2000.

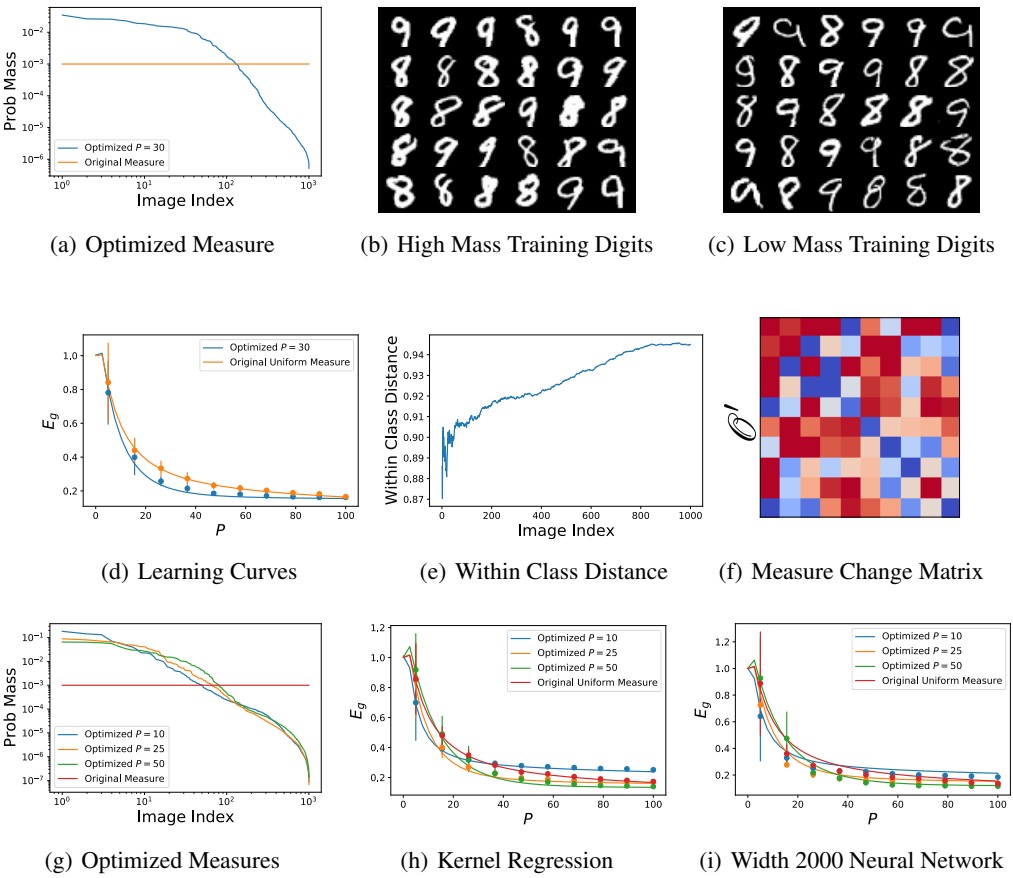

Figure 2: We study regression with a uniform test distribution on MNIST 8's and 9's. For this fixed task, we optimize our theoretical generalization error expression over training distributions at $P = 30$. This gives a probability distribution over MNIST digits shown in (a). The 30 images with highest probability mass (b) appear qualitatively more representative of handwritten 8's and 9's than those given the lowest probability mass (c). We plot the theoretical (solid) and experimental (dots) learning curves for the optimized and original uniform training measure, showing that changing the sampling strategy can improve generalization. Error bars display standard deviation over 30 repeats. (e) After ordering each point by their probability mass on the optimized measure, we calculate the average feature space (in the sense of the kernel) distance to all other points from the same class. This measure rises with image index, indicating that images with higher probability are approximately centroids for each class. (f) The optimized measure induces a non-zero measure change matrix $\mathscr{O}'$. (g) The optimized probability distributions have different shapes for different training budget sizes $P$, with flatter distributions at larger $P$. (h) Kernel regression experiments agree with theory for each of these measures. A measure which performs best at low sample sizes may give sub-optimal generalization at larger sample sizes. (i) The theory also approximates learning curves for finite width 2000 fully connected neural networks with depth 3, initialized with NTK initialization [12].

## 4 Linear Regression: An Analytically Solvable Model

Next we study linear regression to demonstrate various interesting OOD generalization phenomena. Consider a linear target function $\bar{f}(\mathbf{x}) = \beta^\top \mathbf{x}$ where $\mathbf{x}, \beta \in \mathbb{R}^D$ and a linear kernel $K(\mathbf{x}, \mathbf{x}') = \frac{1}{D}\mathbf{x}^\top \mathbf{x}'$ (in this model $M = D$). Data are sampled from zero-mean Gaussians with arbitrary covariance matrices $\mathbf{C}$ and $\tilde{\mathbf{C}}$ for training and test distributions, respectively.

In this case, the kernel eigenvalue equation can be solved exactly. Denoting the eigenvalues and eigenvectors of the covariance matrix of the training distribution as $\mathbf{C}\mathbf{U} = \mathbf{U}\boldsymbol{\Sigma}$ where $\boldsymbol{\Sigma}_{\rho\gamma} = \sigma_\rho^2 \delta_{\rho\gamma}$ is the diagonal eigenvalue matrix and $\mathbf{U}$ is an orthogonal matrix with columns being eigenvectors of

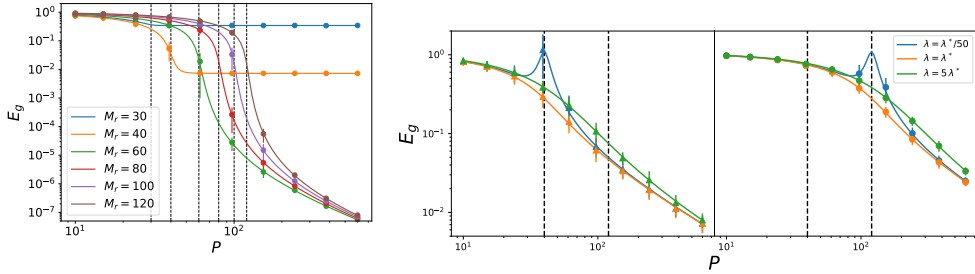

(a) Varying Training Measure Dimension    (b) Effect of Training Measure

Figure 3: (a) Learning curves for linear regression on a target $\bar{f} = \beta^\top \mathbf{x}$ where $\beta_{\rho \leq 40} \sim \mathcal{N}(0,1)$, $\beta_{60 > \rho > 40} \sim \mathcal{N}(0, 0.01)$ and $\beta_{\rho > 60} = 0$, hence $N = 60$. Input dimension, label noise and ridge parameter are $D = 120$, $\varepsilon^2 = 0$ and $\lambda = 10^{-3}$. Altering the training measure dimensionality below or above $M_r = 60$ hurts generalization. For all curves $E_g \to 0$ as $P \to \infty$ except the blue and orange lines at $M_r = 30, 40$ where there is irreducible error since some portion of the target coefficients are not learned. (b) Same experiment with $\varepsilon^2 = 0.1$ label noise and with varying ridge parameters $\lambda$. Target coefficients are $\beta_{\rho \leq 40} \sim \mathcal{N}(0,1)$ and $\beta_{\rho > 40} = 0$ ($N = 40$). The variances of training and test distributions are $\sigma^2 = \tilde{\sigma}^2 = 1$. Left panel and right panels are the learning curves for $M_r = 40, 120$, respectively. Dashed vertical lines indicate the number $P = M_r = 40, 120$. Dots are experiment, lines are theory. Error bars represent standard deviation over 30 trials.

$\mathbf{C}$, the integral eigenvalue problem becomes $\eta_\rho \phi_\rho(\mathbf{x}) = \langle K(\mathbf{x}, .), \phi_\rho(.) \rangle_{p(\mathbf{x})} = \frac{\sigma_\rho}{D} \mathbf{u}_\rho^\top \mathbf{x}$. Therefore, the normalized eigenfunctions are $\phi_\rho(\mathbf{x}) = \mathbf{u}_\rho^\top \mathbf{x} / \sigma_\rho$ and the eigenvalues are $\eta_\rho = \sigma_\rho^2 / D$. The overlap matrix is

$$\mathscr{O}_{\rho\gamma} = \langle \phi_\rho(.), \phi_\gamma(.) \rangle_{\tilde{p}(\mathbf{x})} = \mathbf{\Sigma}^{-1/2} \mathbf{U}^\top \tilde{\mathbf{C}} \mathbf{U} \mathbf{\Sigma}^{-1/2}. \tag{11}$$

Finally computing the target weights as $\mathbf{a} = \mathbf{\Sigma}^{1/2} \mathbf{U}^\top \beta$, we obtain the generalization error Eq.(1):

$$E_g = E_g^{0,p(\mathbf{x})} + \frac{\gamma' - \gamma}{1 - \gamma} \varepsilon^2 + (\kappa D)^2 \beta^\top (P\mathbf{C} + \kappa D\mathbf{I})^{-1} \left( \tilde{\mathbf{C}} - \frac{1 - \gamma'}{1 - \gamma} \mathbf{C} \right) (P\mathbf{C} + \kappa D\mathbf{I})^{-1} \beta,$$

$$\gamma = P \operatorname{Tr} \mathbf{C}^2 (P\mathbf{C} + \kappa D\mathbf{I})^{-2}, \ \gamma' = P \operatorname{Tr} \tilde{\mathbf{C}} \mathbf{C} (P\mathbf{C} + \kappa D\mathbf{I})^{-2},$$

$$\kappa = \lambda + \kappa \operatorname{Tr} \mathbf{C} (P\mathbf{C} + \kappa D\mathbf{I})^{-1} \tag{12}$$

and

$$E_g^{0,p(\mathbf{x})} = \frac{\gamma}{1 - \gamma} \varepsilon^2 + \frac{(\kappa D)^2}{1 - \gamma} \beta^\top \mathbf{C} (P\mathbf{C} + \kappa D\mathbf{I})^{-2} \beta. \tag{13}$$

As a consistency check, we note that the generalization error is minimized ($E_g = 0$) when $\tilde{\mathbf{C}} = 0$ corresponding to a Dirac measure at the origin. This makes sense since target function at origin is $0$ and the estimator on the test distribution is also $0$.

Next, we consider a diagonal covariance matrix $\tilde{\mathbf{C}} = \tilde{\sigma}^2 \mathbf{I}$ for test distribution and

$$\mathbf{C} = \operatorname{diag} (\underbrace{\sigma^2, \dots \sigma^2}_{M_r}, \underbrace{0, \dots 0}_{D - M_r}) \tag{14}$$

for training distribution to demonstrate how training distribution affects generalization in a simplified setting. The integer $M_r$ corresponds to the rank of the training measure's covariance which is also the number of non-zero kernel eigenvalues for the linear kernel. Furthermore we take the target function to have power only in the first $N$ features ($\beta_{\rho > N} = 0$) where we adopt to the normalization $\sum_{\rho=1}^{N} \beta_\rho^2 = 1$. Thus, the target does not depend on the $D - N$ remaining dimensions $x_{\rho > N}$ and we study how compressing some directions in training distribution influences generalization.

In this case, the self-consistent equation for $\kappa$ becomes exactly solvable. The generalization error reduces to:

$$E_g = \tilde{\sigma}^2 \left( \frac{\varepsilon^2}{\sigma^2} \frac{\alpha}{(\kappa' + \alpha)^2 - \alpha} + \frac{\kappa'^2}{(\kappa' + \alpha)^2 - \alpha} \sum_{\rho=1}^{M_r} \beta_\rho^2 + \sum_{\rho=M_r+1}^{N} \beta_\rho^2 \right), \tag{15}$$

where $\kappa' = \frac{\kappa}{\sigma^2 M_r/D} = \frac{1}{2}\left[(1 + \tilde{\lambda} - \alpha) + \sqrt{(1 + \tilde{\lambda} + \alpha)^2 - 4\alpha}\right]$, $\tilde{\lambda} = \frac{\lambda}{\sigma^2 M_r/D}$ and $\alpha = P/M_r$. This result matches that of [16] analyzing in-distribution generalization error when $M_r = N = D$ and $\sigma^2 = \tilde{\sigma}^2$. We identify $\tilde{\lambda}$ as an *effective regularization* parameter as it assumes the role $\lambda$ plays for in-distribution generalization (compare to Eq. 7 of [16]).

First, we note that the generalization linearly scales with the variance of the test distribution. This is due to the linearity of the target and the estimator; any mismatch between them is amplified when the test points are further away from the origin.

Next, when the dimension of training measure is smaller than the dimension of the target function, $M_r < N$, there is an irreducible error due to the fact that kernel machine cannot fit the $N - M_r$ dimensions which are not expressed in the training data (Figure 3a). However in certain cases choosing $M_r < N$ may help generalization if the data budget is limited to $P \sim M_r$ since the learning happens faster compared to larger $M_r$. As illustrated in Figure 3a, if the target power is distributed such that $\sum_{\rho=N'+1}^{N} \beta_\rho^2 \ll \sum_{\rho=1}^{N'} \beta_\rho^2$, choosing $M_r = N'$ can be a better strategy despite the irreducible error due to unexplored target power. In Figure 3a, we picked $N = 60$ and $N' = 40$ which is the number of directions target places most of its power on. If the data budget was limited to $P < N$, choosing $M_r = N'$ (orange curve in Figure 3a) performs better than $M_r = N$ (green curve) although for $P \geq N$ the $M_r < N$ curve has irreduible error while the $M_r = N$ does not. Reducing $M_r$ below $N'$ (blue curve in Figure 3a) on the other hand does not provide much benefit at small $P$.

Next, we observe that $\kappa'$ is a monotonically decreasing function of $\alpha$ and shows a sharp decrease near $\alpha \approx 1 + \tilde{\lambda}$ which also implies a sharp change in generalization error. In fact when $\tilde{\lambda} = 0$, $\kappa'$ becomes zero at $\alpha = 1$ and its first derivative diverges implying either a divergence in generalization error due to label noise or vanishing of $E_g$ in the absence of noise (Section SI.3). When the labels are noisy, this non-monotonic behavior $\alpha = 1$ in $E_g$ (called sample-wise double-descent) signals the over-fitting of the labels beyond which the kernel machine is able to average over noise and converge to the true target function [24, 31, 32, 33, 34, 35, 36, 37, 38, 39, 16]. Hence reducing $M_r$ means that this transition occurs earlier in the learning curve ($P \sim (1 + \tilde{\lambda})M_r$) which implies that less training samples are necessary to estimate the target function well. Intuitively, sampling from an effectively higher dimensional space increases the necessary amount of training data to fit the modes in all directions. In Figure 3(b), we demonstrate this prediction for a linear target and obtain perfect agreement with experiment.

Finally, we show that increasing effective regularization $\tilde{\lambda}$, which is controlled by the training distribution ($\sigma^2$, $M_r$), leads to the suppression of double-descent peak, hence avoiding possible non-monotonicities. However large $\tilde{\lambda}$ leads to slower learning creating a trade-off. Results from previous studies [36, 16] have shown the existence of an optimal ridge parameter in linear regression minimizing the generalization error for all $P$. Minimizing Eq.(15) with respect to $\lambda$, we find that the optimal ridge parameter is given by:

$$\lambda^* = \frac{M_r}{D}\varepsilon^2. \tag{16}$$

In Figure 3(b), we show that choosing optimal ridge parameters indeed mitigates the double descent and results in the best generalization performance.

## 5  Further Results

In SI, we present other analytically solvable models and further analysis on distribution mismatch for real data applications.

- In Section SI.2, we analyze the gradient descent/ascent procedures performed in Section 3 and provide further experiments to motivate how our theory can be used to find mismatched train/test distributions which improve generalization. We also apply our theory to adversarial attacks during testing.
- In Section SI.3, we study a more general linear regression model with diagonal overlap matrix where train/test distributions, target function and number of directions kernel expresses vary. When the target has out-of-RKHS components, we show how distribution shifts may help in generalization.
- In Section SI.4, we apply our theory to rotation invariant kernels such as the Gaussian RBF kernel and NTK [12] acting on spherical data in high-dimensions. We examine how a mismatched sphere radius affects the generalization error in the limit $P, D \to \infty$ similar to the case solved in [15, 16].

- In Section SI.5, we study interpolation versus extrapolation in linear regression and regression in Fourier space by studying rectangular distributions with different ranges for one-dimensional inputs.

## 6 Discussion

Interest in kernel methods has recently surged due to their profound connections to deep neural networks [12, 13, 40, 41], and the replica method of statistical physics proved useful in studying its generalization properties, inductive biases and non-monotonic learning curves [15, 16, 28, 42, 43, 44, 37, 45]. Along the lines of these works, we analyzed generalization performance of kernel regression under distribution mismatch and obtained analytical expressions which fit experiments perfectly including those with wide neural networks. We demonstrated that our formula can be used to optimize over training distribution on MNIST which revealed that certain digits are better to sample more often than others. We considered several analytically solvable models, particularly linear regression for which we demonstrated how dimensional reduction of the training distribution can be helpful in generalization, including shifting of a double-descent peak. Our theory brings many insights about how kernel regression generalizes to unseen distributions.

In this work, we focused on generalization error for specific mismatched training and test distributions, as in [4, 46]. One could, instead, consider a set of possible test distributions that the model could encounter, and assess out-of-distribution generalization error based on the worst performance on that set. This distributional robustness approach [47] has been the focus of most previous studies [19, 48, 49, 50] where one can train a model by minimizing a more general empirical risk on a subset of all test distributions using robust optimization techniques [51]. More recent approaches aim to learn invariant relationships across all distributions [21, 22, 20]. It will be interesting to see if our approach can be generalized to ensembles of training and test distributions.

While our theory accurately describes experiments, it has limitations. First, using it to optimize for a training distribution requires the knowledge of exact test distribution which could not be available at the time of training. Second, the theory requires an eigendecomposition of the kernel which is computationally costly for large datasets. This problem, however, can potentially be solved by using stochastic methods to compute the kernel regression solution [52]. Third, our theory uses the replica theory [14], which is not fully rigorous. Fourth, if to be used to describe neural networks, our theory's applicability is limited to their kernel regime.

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
