# Supplemental Information for "Out-of-Distribution Generalization in Kernel Regression"

## SI.1  Calculation of Generalization Error

### SI.1.1  Problem Formulation

We consider a probability distribution $p(\mathbf{x})$ on the input space $\mathcal{X} \subset \mathbb{R}^D$ and an orthonormal basis $\{\phi_\rho(\mathbf{x})\}|_{\rho=0}^M$ ($M$ is typically infinite) spanning the space of square integrable functions $L^2(\mathcal{X})$ such that any square integrable function can be expanded as:

$$f(\mathbf{x}) = \sum_{\rho=1}^M a_\rho \phi_\rho(\mathbf{x}), \quad \langle f, f \rangle_{p(\mathbf{x})} = \sum_{\rho=1}^M a_\rho^2 < \infty, \tag{SI.1.1}$$

where $\langle f, f \rangle_{p(\mathbf{x})}$ denotes the $L^2$ norm of the function.

A reproducing kernel Hilbert space (RKHS) $\mathcal{H}$ is a Hilbert space endowed with an inner product $\langle ., . \rangle_{\mathcal{H}}$ where evaluation operator is continuous, mapping any function $f \in \mathcal{H}$ to its value at $f(\mathbf{x})$ [26, 53]:

$$f(\mathbf{x}) = \langle f(.), K(., \mathbf{x}) \rangle_{\mathcal{H}} \quad \forall f \in \mathcal{H}. \tag{SI.1.2}$$

Here the so-called reproducing kernel $K : \mathcal{X} \times \mathcal{X} \to \mathbb{R}$ is a positive-definite function whose partial evaluation $K(., \mathbf{x})$ itself belongs to $\mathcal{H}$. It can be characterized by the integral operator $T_K$:

$$[T_K f](x) = \int d\mathbf{x}' p(\mathbf{x}') K(\mathbf{x}, \mathbf{x}') f(\mathbf{x}'), \tag{SI.1.3}$$

where $T_K$ has spectral decomposition $[T_K \phi_\rho](\mathbf{x}) = \eta_\rho \phi_\rho(\mathbf{x})$ for $\rho = 1, \ldots, M$. Then *Mercer's theorem* allows the following kernel representation in terms of orthonormal functions $\{\phi_\rho(\mathbf{x})\}|_{\rho=0}^M$:

$$K(\mathbf{x}, \mathbf{x}') = \sum_{\rho=1}^N \eta_\rho \phi_\rho(\mathbf{x}) \phi_\rho(\mathbf{x}') = \mathbf{\Phi}(\mathbf{x})^\top \mathbf{\Lambda} \mathbf{\Phi}(\mathbf{x}') = \mathbf{\Psi}(\mathbf{x})^\top \mathbf{\Psi}(\mathbf{x}), \tag{SI.1.4}$$

$$\psi_\rho(\mathbf{x}) \equiv \sqrt{\eta_\rho} \phi_\rho(\mathbf{x}), \quad \langle \psi_\rho(\mathbf{x}), \psi_\gamma(\mathbf{x}) \rangle_{\mathcal{H}} = \delta_{\rho\gamma},$$

where we defined the diagonal eigenvalue matrix $\mathbf{\Lambda}_{\rho\gamma} = \eta_\rho \delta_{\rho\gamma}$, $M$-dimensional vector $\left( \mathbf{\Phi}(\mathbf{x}) \right)_\rho = \phi_\rho(\mathbf{x})$ and features $\psi_\rho$. With this representation Hilbert inner product $\langle ., . \rangle_{\mathcal{H}}$ reduces to:

$$\langle f, g \rangle_{\mathcal{H}} \equiv \sum_{\rho=1}^M \frac{a_\rho b_\rho}{\eta_\rho}, \tag{SI.1.5}$$

for two functions $f(\mathbf{x}) = \mathbf{a}^\top \mathbf{\Phi}(\mathbf{x})$ and $g(\mathbf{x}) = \mathbf{b}^\top \mathbf{\Phi}(\mathbf{x})$. A function belongs to this RKHS only if its Hilbert norm is finite:

$$\|f\|_{\mathcal{H}}^2 \equiv \langle f, f \rangle_{\mathcal{H}} = \sum_{\rho=1}^M \frac{a_\rho^2}{\eta_\rho} < \infty. \tag{SI.1.6}$$

Note that the kernel does not have to represent all $M$-features meaning that its eigenvalues may truncate at some integer $N < M$: $\eta_{\rho > N} = 0$. If this is the case, then functions which have power on modes $\rho > N$ are out-of-RKHS functions since they have infinite Hilbert norm.

Given a finite training data $\mathcal{D} = \{\mathbf{x}^\mu, y^\mu\}_{\mu=1}^P$ where inputs are drawn from probability distribution $p(\mathbf{x})$, we wish to study kernel regression on the RKHS $\mathcal{H}$. First, we assume that the labels are generated by a target function $\bar{f}(\mathbf{x})$ with additive noise:

$$y^\mu = \bar{f}(\mathbf{x}^\mu) + \epsilon^\mu, \tag{SI.1.7}$$

where $\mathbb{E}[\epsilon^\mu \epsilon^\nu] = \varepsilon^2 \delta_{\mu\nu}$ are independent for each training sample. In general, the target function does not have to be in the RKHS and the out-of-RKHS components need to be treated separately when

$\eta_\rho = 0$ for some $\rho$. However we find that taking the limit $\eta_\rho \to 0$ at the end of the calculation yields the correct expressions for out-of-RKHS cases hence we keep all $\eta_\rho \neq 0$ for now. Expanding the target function in terms of the eigenfunctions with respect to $p(\mathbf{x})$:

$$y^\mu = \bar{\mathbf{a}}^\top \mathbf{\Phi}(\mathbf{x}) + \epsilon^\mu = \bar{\mathbf{w}}^\top \mathbf{\Psi}(\mathbf{x}^\mu) + \epsilon^\mu. \tag{SI.1.8}$$

The problem of interest is the minimization of the energy function $H[f; \mathcal{D}]$ with respect to functions $f \in \mathcal{H}$:

$$\mathbf{w}^* = \text{argmin}_{\mathbf{w} \in \mathbf{R}^M} H(\mathbf{w}; \mathcal{D}), \quad H(\mathbf{w}; \mathcal{D}) \equiv \frac{1}{2\lambda} \sum_{\mu=1}^{P} \left( \mathbf{\Psi}(\mathbf{x}^\mu) \cdot (\bar{\mathbf{w}} - \mathbf{w}) + \epsilon^\mu \right)^2 + \frac{1}{2} \|\mathbf{w}\|_2^2, \tag{SI.1.9}$$

Then the resulting estimator becomes:

$$f^*(\mathbf{x}; \mathcal{D}) = \mathbf{w}^{*\top} \mathbf{\Psi}(\mathbf{x}), \tag{SI.1.10}$$

Note that the estimator is always in the RKHS, meaning it should not depend on eigenfunctions $\phi_\rho(\mathbf{x})$ if the corresponding eigenvalue $\eta_\rho = 0$.

The estimator above depends on the particular choice of a training set and it is difficult to obtain an analytical expression for it. Hence we would like to compute the average case estimator which only depends on the input distribution, size of the dataset, target function and the hypothesis class $\mathcal{H}$ but not the individual training samples. Next we discuss how to perform dataset averaging for kernel regression using methods from statistical physics.

## SI.1.2 Replica Calculation for Generalization

We would like to calculate the observables $\mathcal{O}[f^*]$ of the estimator $f^*(\mathbf{x})$ averaged over the training dataset $\mathcal{D}$. These observables include the generalization and training errors. For our purposes, we only need to calculate the mean and variance of the estimator which completely determines the generalization error. To perform this calculation, we introduce the following partition function:

$$Z[\boldsymbol{\xi}, \boldsymbol{\chi}] = \int d\mathbf{w} e^{-\beta H(\mathbf{w}; \mathcal{D}) + \beta \boldsymbol{\xi} \cdot \mathbf{w} + \frac{\beta}{2} \boldsymbol{\chi}^\top \mathbf{w} \mathbf{w}^\top \boldsymbol{\chi}}, \tag{SI.1.11}$$

where $\beta$ is inverse temperature and $\boldsymbol{\xi}, \boldsymbol{\chi}$ are source terms to compute expectation values of the estimator weights. The partition function represents a probability distribution over all possible estimators $\mathbf{w}$ and as $\beta \to \infty$ it concentrates around the kernel regression solution $\mathbf{w}^*$.

We can calculate the dataset averaged estimator $f^*(\mathbf{x})$ and its correlation function via:

$$\sqrt{\eta_\alpha} \mathbb{E}_{\mathcal{D}} w_\alpha^*(\mathcal{D}) = \lim_{\beta \to \infty} \frac{1}{\beta} \frac{\partial}{\partial \xi_\alpha'} \mathbb{E}_{\mathcal{D}} \log Z[\boldsymbol{\xi}, \boldsymbol{\chi}] \Big|_{\boldsymbol{\xi}, \boldsymbol{\chi}=0}$$

$$\sqrt{\eta_\alpha \eta_\beta} \mathbb{E}_{\mathcal{D}}[w_\alpha^*(\mathcal{D}) w_\beta^*(\mathcal{D})] = \lim_{\beta \to \infty} \frac{1}{\beta} \frac{\partial^2}{\partial \chi_\alpha' \partial \chi_\beta'} \mathbb{E}_{\mathcal{D}} \log Z[\boldsymbol{\xi}, \boldsymbol{\chi}] \Big|_{\boldsymbol{\xi}, \boldsymbol{\chi}=0}, \tag{SI.1.12}$$

where the primed quantities are defined as $\boldsymbol{\xi} = \mathbf{\Lambda}^{1/2} \boldsymbol{\xi}'$ and $\boldsymbol{\chi} = \mathbf{\Lambda}^{1/2} \boldsymbol{\chi}'$. Hence one can read out:

$$\mathbb{E}_{\mathcal{D}} f^*(\mathbf{x}, \mathcal{D}) = \sum_\alpha \mathbb{E}_{\mathcal{D}} w_\alpha^*(\mathcal{D}) \sqrt{\eta_\alpha} \phi_\alpha(\mathbf{x})$$

$$\mathbb{E}_{\mathcal{D}}[f^*(\mathbf{x}, \mathcal{D}) f^*(\mathbf{x}', \mathcal{D})] = \sum_{\alpha\beta} \mathbb{E}_{\mathcal{D}}[w_\alpha^*(\mathcal{D}) w_\beta^*(\mathcal{D})] \sqrt{\eta_\alpha \eta_\beta} \phi_\alpha(\mathbf{x}) \phi_\beta(\mathbf{x}'). \tag{SI.1.13}$$

However, computing the average of $\log Z$ over all possible training samples and noises is a challenging task due to the integrals of logarithms. This is where we resort to replica trick which replaces averaging $\log Z$ with averaging $Z^n$, n-times *replicated* partition function:

$$\mathbb{E}_{\mathcal{D}} \log Z = \lim_{n \to 0} \frac{\mathbb{E}_{\mathcal{D}} Z^n - 1}{n}. \tag{SI.1.14}$$

The calculation of $\mathbb{E}_{\mathcal{D}} Z^n$ is done for integer $n$ and then analytically continued to real numbers to perform the limit. Despite being non-rigorous, the replica method has proven powerful and predictive in the study of disordered systems as well as neural networks and machine learning (see [14, 54, 27] for reviews).

Plugging in the eigenfunction expansions derived above, $Z^n$ becomes:

$$\mathbb{E}_{\mathcal{D}} Z^n = e^{-\frac{n\beta}{2}(\bar{\mathbf{W}}-\boldsymbol{\xi})^{\top}\bar{\mathbf{w}}} \int \left( \prod_{a=1}^{n} d\mathbf{w}^a \right) e^{-\frac{\beta}{2} \sum_{a=1}^{n} \mathbf{w}^{a\top}(\mathbf{I}-\boldsymbol{\chi}\boldsymbol{\chi}^{\top})\mathbf{w}^a - \beta \bar{\mathbf{W}}^{\top} \sum_{a=1}^{n} \mathbf{w}^a}$$
$$\times \left\langle e^{-\frac{\beta}{2\lambda} \sum_{a=1}^{n} \left( \mathbf{w}^a \cdot \boldsymbol{\Psi}(\mathbf{x}) + \epsilon^a \right)^2} \right\rangle_{\mathbf{x} \sim p(\mathbf{x}), \{\epsilon^a\}}^{P}, \quad \text{(SI.1.15)}$$

where we shifted $\mathbf{w}^a \to \mathbf{w}^a + \bar{\mathbf{w}}$ and defined $\bar{\mathbf{W}} = (\bar{\mathbf{w}} - \boldsymbol{\xi}) - (\boldsymbol{\chi}^{\top}\bar{\mathbf{w}})\boldsymbol{\chi}$ for notational convenience. A crucial step is to compute the expectation value over possible realizations of training sets which behave as quenched disorders in our system. Our strategy to perform this quenched average is approximating the exponent as a Gaussian random variable defined by $q^a = (\mathbf{w}^a - \bar{\mathbf{w}}) \cdot \boldsymbol{\Psi}(\mathbf{x}) + \epsilon^a$ whose second-order statistics is given by:

$$\boldsymbol{\mu}^a \equiv \langle q^a \rangle = 0,$$
$$\mathbf{C}^{ab} \equiv \langle q^a q^b \rangle = (\mathbf{w}^a - \bar{\mathbf{w}})^{\top} \langle \boldsymbol{\Psi}(\mathbf{x})\boldsymbol{\Psi}(\mathbf{x})^T \rangle (\mathbf{w}^b - \bar{\mathbf{w}}) + \langle \epsilon^a \epsilon^b \rangle = (\mathbf{w}^a - \bar{\mathbf{w}})^{\top} \boldsymbol{\Lambda} (\mathbf{w}^b - \bar{\mathbf{w}}) + \boldsymbol{\Sigma}^{ab},$$
$$\text{(SI.1.16)}$$

where $\boldsymbol{\Sigma} = (\varepsilon^2 + \|\bar{\mathbf{a}}\|_2^2)\mathbf{1}\mathbf{1}^{\top}$ is the covariance matrix of noise across replicas. We assumed that the kernel does not include a constant mode so that $\boldsymbol{\mu}^a = 0$. Noticing that $q^a$ is a summation of many uncorrelated random variables ($\langle \psi_{\rho}(\mathbf{x}), \psi_{\rho'}(\mathbf{x}) \rangle_{p(\mathbf{x})} = \eta_{\rho} \delta_{\rho\rho'}$) and a Gaussian noise, we approximate the probability distribution of $q^a$ by a multivariate Gaussian with its mean and covariance given by Eq.(SI.1.16):

$$P(\{q^a\}) = \frac{1}{\sqrt{(2\pi)^n \det(\mathbf{C})}} \exp\left( -\frac{1}{2} \sum_{a,b} q^a (\mathbf{C}^{ab})^{-1} q^b \right), \quad \text{(SI.1.17)}$$

The Gaussian approximation proves accurate given the excellent match of our theory to simulations. This reduces the average over quenched disorder to:

$$\left\langle e^{-\frac{\beta}{2\lambda} \sum_{a=1}^{n} \left( (\mathbf{w}^a - \bar{\mathbf{w}}) \cdot \boldsymbol{\Psi}(\mathbf{x}) + \epsilon^a \right)^2} \right\rangle_{\mathbf{x}, \{\epsilon^a\}} \approx \int \{dq^a\} P(\{q^a\}) \exp\left( -\frac{\beta}{2\lambda} \sum_{a=1}^{n} (q^a)^2 \right)$$
$$= \exp\left( -\frac{1}{2} \log \det\left( \mathbf{I} + \frac{\beta}{\lambda}\mathbf{C} \right) \right). \quad \text{(SI.1.18)}$$

Combining everything together, the averaged replicated partition function becomes:

$$\mathbb{E}_{\mathcal{D}} Z^n = e^{-\frac{n\beta}{2}(\bar{\mathbf{W}}-\boldsymbol{\xi})^{\top}\bar{\mathbf{w}}} \int \left( \prod_{a=1}^{n} d\mathbf{w}^a \right) e^{-\frac{\beta}{2} \sum_{a=1}^{n} \mathbf{w}^{a\top}(\mathbf{I}-\boldsymbol{\chi}\boldsymbol{\chi}^{\top})\mathbf{w}^a - \beta \bar{\mathbf{W}}^{\top} \sum_{a=1}^{n} \mathbf{w}^a - \frac{P}{2} \log \det\left( \mathbf{I} + \frac{\beta}{\lambda}\mathbf{C} \right)}, \quad \text{(SI.1.19)}$$

Using the definitions Eq.(SI.1.16), we insert the following identity to the integral:

$$1 = \left( \frac{iP}{2\pi} \right)^{\frac{n(n+1)}{2}} \int \left( \prod_{a \geq b} d\mathbf{C}^{ab} d\hat{\mathbf{C}}^{ab} \right) \exp\left[ -P \sum_{a \geq b} \hat{\mathbf{C}}^{ab} \left( \mathbf{C}^{ab} - \mathbf{w}^{a\top} \boldsymbol{\Lambda} \mathbf{w}^b - \boldsymbol{\Sigma}^{ab} \right) \right] \quad \text{(SI.1.20)}$$

Here, the integral over $\hat{\mathbf{C}}$ runs over the imaginary axis and we explicitly scaled conjugate variables by $P$. Then defining:

$$G_E = \frac{1}{2} \log \det\left( \mathbf{I} + \frac{\beta}{\lambda}\mathbf{C} \right), \quad \text{(SI.1.21)}$$

$$e^{-G_S} = \int \left( \prod_{a=1}^{n} d\mathbf{w}^a \right) \exp\left( -\frac{\beta}{2} \sum_{a \geq b} \mathbf{w}^{a\top} \left( (\mathbf{I} - \boldsymbol{\chi}\boldsymbol{\chi}^{\top})\mathbf{I}^{ab} - \frac{2P}{\beta} \boldsymbol{\Lambda} \hat{\mathbf{C}}^{ab} \right) \mathbf{w}^b - \beta \bar{\mathbf{W}}^{\top} \sum_{a=1}^{n} \mathbf{w}^a \right), \quad \text{(SI.1.22)}$$

we obtain:

$$\mathbb{E}_{\mathcal{D}} Z^n = e^{\frac{n(n+1)}{2}\log\left(\frac{iP}{2\pi}\right)-\frac{n\beta}{2}(\bar{\mathbf{W}}-\boldsymbol{\xi})^\top\bar{\mathbf{w}}} \int \left(\prod_{a\geq b} d\mathbf{C}^{ab} d\hat{\mathbf{C}}^{ab}\right) \exp\left[-P\sum_{a\geq b}\hat{\mathbf{C}}^{ab}(\mathbf{C}^{ab}-\boldsymbol{\Sigma}^{ab})-PG_E-G_S\right].$$

(SI.1.23)

Next, we need to evaluate the integral in $G_S$. First we would like to express the ordered sum $\sum_{a\geq b}\mathbf{w}^{a\top}\left((\mathbf{I}-\boldsymbol{\chi}\boldsymbol{\chi}^\top)\mathbf{I}^{ab}-\frac{2P}{\beta}\boldsymbol{\Lambda}\hat{\mathbf{C}}^{ab}\right)\mathbf{w}^b$ as an unordered sum over $a,b$. Note that

$$\sum_{a,b}\mathbf{w}^{a\top}\left((\mathbf{I}-\boldsymbol{\chi}\boldsymbol{\chi}^\top)\mathbf{I}^{ab}-\frac{2P}{\beta}\boldsymbol{\Lambda}\hat{\mathbf{C}}^{ab}\right)\mathbf{w}^b$$

$$=2\sum_{a\geq b}\mathbf{w}^{a\top}\left((\mathbf{I}-\boldsymbol{\chi}\boldsymbol{\chi}^\top)\mathbf{I}^{ab}-\frac{2P}{\beta}\boldsymbol{\Lambda}\hat{\mathbf{C}}^{ab}\right)\mathbf{w}^b-\sum_{a,b}\mathbf{w}^{a\top}\left((\mathbf{I}-\boldsymbol{\chi}\boldsymbol{\chi}^\top)\mathbf{I}^{ab}-\frac{2P}{\beta}\boldsymbol{\Lambda}\text{diag}(\hat{\mathbf{C}})^{ab}\right)\mathbf{w}^b$$

Hence, we obtain:

$$\sum_{a\geq b}\mathbf{w}^{a\top}\left((\mathbf{I}-\boldsymbol{\chi}\boldsymbol{\chi}^\top)\mathbf{I}^{ab}-\frac{2P}{\beta}\boldsymbol{\Lambda}\hat{\mathbf{C}}^{ab}\right)\mathbf{w}^b=\sum_{a,b}\mathbf{w}^{a\top}\mathbf{X}^{ab}\mathbf{w}^b,$$

where we defined:

$$\mathbf{X}^{ab}=\left(\mathbf{I}-\boldsymbol{\chi}\boldsymbol{\chi}^\top\right)\mathbf{I}^{ab}-\frac{P}{\beta}\boldsymbol{\Lambda}\left(\hat{\mathbf{C}}+\text{diag}(\hat{\mathbf{C}})\right)^{ab}.$$

(SI.1.24)

In order to evaluate the Gaussian integral, we will cast the function and target weights into an $nM$ dimensional vector:

$$\mathbf{w}=\begin{pmatrix}\mathbf{w}^1 & \mathbf{w}^2 & .. & \mathbf{w}^a & .. & \mathbf{w}^n\end{pmatrix}_{nM\times 1}$$
$$\bar{\mathbf{W}}_{\otimes n}=\begin{pmatrix}\bar{\mathbf{W}} & \bar{\mathbf{W}} & \ldots & \bar{\mathbf{W}}\end{pmatrix}_{nM\times 1}$$

(SI.1.25)

Furthermore, we introduce the $nM\times nM$ matrix $\mathbf{X}$ as:

$$\mathbf{X}=\begin{pmatrix}\mathbf{X}^{11} & \mathbf{X}^{12} & \ldots & \ldots & \mathbf{X}^{1n}\\ \mathbf{X}^{21} & \mathbf{X}^{22} & \ldots & \ldots & \mathbf{X}^{2n}\\ \vdots & \vdots & \ddots & & \vdots\\ \vdots & \ldots & \mathbf{X}^{ab} & \ddots & \vdots\\ \mathbf{X}^{n1} & \ldots & \ldots & \ldots & \mathbf{X}^{nn}\end{pmatrix}_{nM\times nM}$$

(SI.1.26)

Finally, we denote the integration measure as $\mathcal{D}\mathbf{w}=\prod_{a,\rho}dw_\rho^a$. With these definitions, $G_S$ becomes:

$$e^{-G_S}=\int\mathcal{D}\mathbf{w}\,e^{-\frac{\beta}{2}\mathbf{w}^\top\mathbf{X}\mathbf{w}-\beta\bar{\mathbf{W}}_{\otimes n}^\top\mathbf{w}}$$

(SI.1.27)

Hence, we turned the integral in $G_S$ to a simple Gaussian integral. The result is:

$$e^{-G_S}=\left(\frac{2\pi}{\beta}\right)^{\frac{nM}{2}}\left(\det\mathbf{X}\right)^{-\frac{1}{2}}\exp\left(\frac{\beta}{2}\bar{\mathbf{W}}_{\otimes n}^\top\mathbf{X}^{-1}\bar{\mathbf{W}}_{\otimes n}\right).$$

(SI.1.28)

Now the integral in Eq.(SI.1.23) can be evaluated using the method of steepest descent. In Eq.(SI.1.23), we see that all the terms in the exponent is $\mathcal{O}(n)$. Furthermore, we will use $P$ as the saddle point parameter going to infinity with proper scaling. Therefore, defining the following function:

$$S[\mathbf{C}, \hat{\mathbf{C}}, \boldsymbol{\mu}, \hat{\boldsymbol{\mu}}] = \frac{1}{n}\sum_{a \geq b}\hat{\mathbf{C}}^{ab}(\mathbf{C}^{ab} - \boldsymbol{\Sigma}^{ab}) + \frac{1}{nP}\left(PG_E + G_S + \frac{n\beta}{2}(\bar{\mathbf{W}} - \boldsymbol{\xi})^{\top}\bar{\mathbf{w}}\right)$$

$$G_E = \frac{1}{2}\log\det\left(\mathbf{I} + \frac{\beta_K}{\lambda}\mathbf{C}\right)$$

$$G_S = \frac{1}{2}\log\det\mathbf{X} - \frac{\beta}{2}\bar{\mathbf{W}}_{\otimes n}^{\top}\mathbf{X}^{-1}\bar{\mathbf{W}}_{\otimes n}, \tag{SI.1.29}$$

we obtain:

$$\mathbb{E}_{\mathcal{D}}\log Z = \lim_{n\to 0}\frac{1}{n}\left(\mathbb{E}_{\mathcal{D}}Z^n - 1\right),$$

$$\mathbb{E}_{\mathcal{D}}Z^n = e^{\frac{n(n+1)}{2}\log\left(\frac{iP}{2\pi}\right) + \frac{nM}{2}\log\frac{2\pi}{\beta}}\int\left(\prod_{a\geq b}d\mathbf{C}^{ab}d\hat{\mathbf{C}}^{ab}\right)e^{-nPS[\mathbf{C},\hat{\mathbf{C}}]}. \tag{SI.1.30}$$

The reader may question the dependence of various quantities in $S$ on $P$, since we are taking a $P \to \infty$ limit. This is because we want to keep our treatment general. Depending on the kernel and data distribution, there are other quantities here that can scale with $P$. Specific examples will be given.

### SI.1.3 Replica Symmetry and Saddle Point Equations

To proceed with the saddle point integration, we further assume replica symmetry relying on the convexity of the problem:

$$C^0 = \mathbf{C}^{aa}, \qquad\qquad \hat{C}^0 = \hat{\mathbf{C}}^{aa},$$
$$C = \mathbf{C}^{a\neq b}, \qquad\qquad \hat{C} = \hat{\mathbf{C}}^{a\neq b}. \tag{SI.1.31}$$

Therefore, we have $\mathbf{C} = (C_0 - C)\mathbf{I} + C\mathbf{1}\mathbf{1}^{\top}$ and $\hat{\mathbf{C}} = (\hat{C}_0 - \hat{C})\mathbf{I} + \hat{C}\mathbf{1}\mathbf{1}^{\top}$. In this case, the matrix $\mathbf{X}$ has the form:

$$\mathbf{X} = \begin{pmatrix}\mathbf{X}_0 & \mathbf{X}_1 & \mathbf{X}_1 & \ldots & \mathbf{X}_1 \\ \mathbf{X}_1 & \mathbf{X}_0 & \mathbf{X}_1 & \ldots & \mathbf{X}_1 \\ \mathbf{X}_1 & \mathbf{X}_1 & \mathbf{X}_0 & \ldots & \vdots \\ \vdots & \ldots & \ldots & \ddots & \vdots \\ \mathbf{X}_1 & \ldots & \ldots & \ldots & \mathbf{X}_0\end{pmatrix} = \mathbf{I}_{n\times n}\otimes(\mathbf{X}_0 - \mathbf{X}_1)_{M\times M} + \mathbf{1}_{n\times n}\otimes(\mathbf{X}_1)_{M\times M}, \tag{SI.1.32}$$

where:

$$\mathbf{X}_0 \equiv \mathbf{X}^{aa} = \left(\mathbf{I} - \boldsymbol{\chi}\boldsymbol{\chi}^{\top}\right) - \frac{2P\hat{C}_0}{\beta}\boldsymbol{\Lambda}$$

$$\mathbf{X}_1 \equiv \mathbf{X}^{a\neq b} = -\frac{P\hat{C}}{\beta}\boldsymbol{\Lambda}.$$

It is straightforward to calculate the inverse of this matrix using Sherman-Morrison-Woodbury formula $(A + B)^{-1} = A^{-1} - A^{-1}BA^{-1}(I + BA^{-1})^{-1}$:

$$\mathbf{X}^{-1} = \mathbf{I}_n\otimes(\mathbf{X}_0 - \mathbf{X}_1)^{-1} - \left(\mathbf{1}_n\otimes(\mathbf{X}_0 - \mathbf{X}_1)^{-1}\mathbf{X}_1(\mathbf{X}_0 - \mathbf{X}_1)^{-1}\right)\left(\mathbf{I}_n\otimes\mathbf{I}_M + \mathbf{1}_n\otimes\mathbf{X}_1(\mathbf{X}_0 - \mathbf{X}_1)^{-1}\right)^{-1}$$
$$= \mathbf{I}_n\otimes\mathbf{Q}^{-1} - \mathbf{1}_n\otimes\mathbf{Q}^{-1}\mathbf{X}_1\mathbf{Q}^{-1} + \mathcal{O}(n),$$

where we defined,

$$\mathbf{Q} \equiv \mathbf{X}_0 - \mathbf{X}_1 = \mathbf{I} - \frac{P(2\hat{C}_0 - \hat{C})}{\beta}\mathbf{\Lambda} - \boldsymbol{\chi}\boldsymbol{\chi}^\top, \qquad \text{(SI.1.33)}$$

for shorthand notation. Hence, we get:

$$\bar{\mathbf{W}}_{\otimes n}^\top \mathbf{X}^{-1} \bar{\mathbf{W}}_{\otimes n} = n\bar{\mathbf{W}}^\top \mathbf{Q}^{-1}\bar{\mathbf{W}} + \mathcal{O}(n^2). \qquad \text{(SI.1.34)}$$

We also need to calculate the determinant of this matrix which can be done by using Gaussian elimination method to bring it into a block-triangular form. The result is:

$$\det \mathbf{X} = \det(\mathbf{X}_0 - \mathbf{X}_1)^{n-1} \det\big(\mathbf{X}_0 + (n-1)\mathbf{X}_1\big) = \det(\mathbf{X}_0 - \mathbf{X}_1)^{n-1} \det(\mathbf{X}_0 - \mathbf{X}_1 + n\mathbf{X}_1). \tag{SI.1.35}$$

Taylor expanding the last term using $\det(\mathbf{I} + n\mathbf{C}) = 1 + n\operatorname{Tr}\mathbf{C} + \mathcal{O}(n^2)$, we obtain:

$$\log \det \mathbf{X} = n\log \det \mathbf{Q} + n\operatorname{Tr}\Big(\mathbf{X}_1\mathbf{Q}^{-1}\Big) + \mathcal{O}(n^2) = n\log \det \mathbf{Q} - n\frac{P\hat{C}}{\beta}\operatorname{Tr}\mathbf{\Lambda}\mathbf{Q}^{-1} + \mathcal{O}(n^2). \tag{SI.1.36}$$

Next, using the matrix determinant lemma $\det\big(A + uv^T\big) = \det(A)(1 + v^T A^{-1}u)$, we obtain:

$$\det\left(\mathbf{I} + \frac{\beta}{\lambda}\mathbf{C}\right) = \Big[1 + \frac{\beta}{\lambda}(C_0 - C)\Big]^n \left(1 + n\frac{\beta C}{\lambda + \beta(C_0 - C)}\right),$$
$$\Rightarrow \log \det\left(\mathbf{I} + \frac{\beta}{\lambda}\mathbf{C}\right) = n\log\left(1 + \frac{\beta}{\lambda}(C_0 - C)\right) + n\frac{\beta C}{\lambda + \beta(C_0 - C)}, \tag{SI.1.37}$$

Finally, we need to simplify $\sum_{a \geq b} \hat{\mathbf{C}}^{ab}(\mathbf{C}^{ab} - \mathbf{\Sigma}^{ab})$ under the replica symmetry up to leading order in $n$:

$$\sum_{a \geq b} \hat{\mathbf{C}}^{ab}(\mathbf{C}^{ab} - \mathbf{\Sigma}^{ab}) = n\big(\hat{C}_0(C_0 - \varepsilon^2) - \frac{1}{2}\hat{C}(C - \varepsilon^2)\big). \tag{SI.1.38}$$

Therefore, under replica symmetry, the function $S$ given in Eq.(SI.1.29) simplifies to:

$$S[\mathbf{C}, \hat{\mathbf{C}}] = \hat{C}_0(C_0 - \varepsilon^2) - \frac{1}{2}\hat{C}(C - \varepsilon^2) + \frac{1}{2}\log\left(1 + \frac{\beta}{\lambda}(C_0 - C)\right) + \frac{1}{2}\frac{\beta C}{\lambda + \beta(C_0 - C)}$$
$$+ \frac{1}{2P}\left(\log \det \mathbf{Q} - \frac{P\hat{C}}{\beta}\operatorname{Tr}\mathbf{\Lambda}\mathbf{Q}^{-1}\right) - \frac{\beta}{2P}\bar{\mathbf{W}}^\top \mathbf{Q}^{-1}\bar{\mathbf{W}} + \frac{\beta}{2P}(\bar{\mathbf{W}} - \boldsymbol{\xi})^\top \bar{\mathbf{w}}, \tag{SI.1.39}$$

where we recall that $\mathbf{Q} = \mathbf{I} - \frac{P(2\hat{C}_0 - \hat{C})}{\beta}\mathbf{\Lambda} - \boldsymbol{\chi}\boldsymbol{\chi}^\top$. The saddle point equations of $S$ with respect to $C_0$ and $C$ are simple:

$$\frac{\partial S}{\partial C} = 0 \Rightarrow \boxed{\hat{C} = \frac{\beta^2 C}{\big(\lambda + \beta(C_0 - C)\big)^2}},$$

$$\frac{\partial S}{\partial C_0} = 0 \Rightarrow \boxed{\hat{C}_0 = \frac{1}{2}\hat{C} - \frac{1}{2}\frac{\beta}{\lambda + \beta(C_0 - C)}}. \tag{SI.1.40}$$

The equation $\partial S/\partial \hat{C} = 0$ yields:

$$C = \frac{P\hat{C}}{\beta^2} \operatorname{Tr} \mathbf{\Lambda}\mathbf{Q}^{-1}\mathbf{\Lambda}\mathbf{Q}^{-1} + \bar{\mathbf{W}}^\top \mathbf{Q}^{-1}\mathbf{\Lambda}\mathbf{Q}^{-1}\bar{\mathbf{W}} + \varepsilon^2, \tag{SI.1.41}$$

and the equation $\partial S/\partial \hat{C}_0 = 0$ yields:

$$C_0 = C + \frac{1}{\beta} \operatorname{Tr} \mathbf{\Lambda}\mathbf{Q}^{-1}. \tag{SI.1.42}$$

Two commonly appearing forms are:

$$\kappa \equiv \lambda + \beta(C_0 - C) = \lambda + \operatorname{Tr} \mathbf{\Lambda}\mathbf{Q}^{-1},$$
$$\frac{2\hat{C}_0 - \hat{C}}{\beta} = -\frac{1}{\lambda + \beta(C_0 - C)} = -\frac{1}{\kappa}. \tag{SI.1.43}$$

Plugging second equation to the expression for $\mathbf{G}$, we get:

$$\mathbf{Q} = \mathbf{I} + \frac{P}{\kappa}\mathbf{\Lambda} - \boldsymbol{\chi}\boldsymbol{\chi}^\top, \tag{SI.1.44}$$

hence we obtain the following implicit equation:

$$\kappa = \lambda + \operatorname{Tr} \mathbf{\Lambda}\left(\mathbf{I} + \frac{P}{\kappa}\mathbf{\Lambda} - \boldsymbol{\chi}\boldsymbol{\chi}^\top\right)^{-1}. \tag{SI.1.45}$$

In terms of $\kappa$, final saddle point equations reduce to:

$$\hat{C}_0^* = \frac{1}{2}\hat{C}^* - \frac{1}{2}\frac{\beta}{\kappa},$$
$$\hat{C}^* = \frac{\beta^2 C^*}{\kappa^2},$$
$$C_0^* = C^* + \frac{\kappa - \lambda}{\beta},$$
$$C^* = C^*\frac{P}{\kappa^2} \operatorname{Tr} \mathbf{\Lambda}\mathbf{Q}^{-1}\mathbf{\Lambda}\mathbf{Q}^{-1} + \bar{\mathbf{W}}^\top \mathbf{Q}^{-1}\mathbf{\Lambda}\mathbf{Q}^{-1}\bar{\mathbf{W}} + \varepsilon^2. \tag{SI.1.46}$$

Here, $^*$ indicates the quantities that give the saddle point. Finally, solving for $C^*$ in the last equation, we obtain:

$$C^* = \frac{1}{1 - \frac{P}{\kappa^2} \operatorname{Tr} \mathbf{\Lambda}\mathbf{Q}^{-1}\mathbf{\Lambda}\mathbf{Q}^{-1}}\left(\bar{\mathbf{W}}^\top \mathbf{Q}^{-1}\mathbf{\Lambda}\mathbf{Q}^{-1}\bar{\mathbf{W}} + \varepsilon^2\right). \tag{SI.1.47}$$

Having obtained the saddle points, we can evaluate the saddle point integral. In the limit $P \to \infty$, the dominant contribution is:

$$\mathbb{E}_{\mathcal{D}}Z^n \approx e^{-nPS[\mathbf{C}^*, \hat{\mathbf{C}}^*]}. \tag{SI.1.48}$$

Taking the $n \to 0$ limit and plugging in the saddle point solutions to the expression Eq.(SI.1.39), we obtain the free energy $\mathbb{E}_{\mathcal{D}}\log Z = -PS$ to be:

$$\mathbb{E}_{\mathcal{D}}\log Z = \frac{P}{2}\frac{\kappa - \lambda}{\kappa} - \frac{P}{2}\log\frac{\kappa}{\lambda} - \frac{P}{2}\log\det\mathbf{Q} - \frac{\beta P}{2}\frac{\varepsilon^2}{\kappa} + \frac{\beta}{2}\bar{\mathbf{W}}^\top\mathbf{Q}^{-1}\bar{\mathbf{W}} - \frac{\beta}{2}(\bar{\mathbf{W}} - \boldsymbol{\xi})^\top\bar{\mathbf{w}},$$
$$\kappa = \lambda + \operatorname{Tr} \mathbf{\Lambda}\mathbf{Q}^{-1},$$
$$\mathbf{Q} = \mathbf{I} + \frac{P}{\kappa}\mathbf{\Lambda} - \boldsymbol{\chi}\boldsymbol{\chi}^\top,$$
$$\bar{\mathbf{W}} = (\bar{\mathbf{w}} - \boldsymbol{\xi}) - (\boldsymbol{\chi}^\top\bar{\mathbf{w}})\boldsymbol{\chi}. \tag{SI.1.49}$$

### SI.1.4   Expected Estimator and the Correlation Function

Finally, we can calculate the RKHS weights of the expected function and its variance:

$$\sqrt{\eta_\alpha}\mathbb{E}_{\mathcal{D}}w_\alpha^* = \lim_{\beta\to\infty}\frac{1}{\beta}\frac{\partial}{\partial\xi_\alpha'}\mathbb{E}_{\mathcal{D}}\log Z\bigg|_{\xi,\chi=0}$$

$$\sqrt{\eta_\alpha\eta_\beta}\mathbb{E}_{\mathcal{D}}[w_\alpha^*(\mathcal{D})w_\beta^*(\mathcal{D})] = \lim_{\beta\to\infty}\frac{1}{\beta}\frac{\partial^2}{\partial\chi_\alpha'\partial\chi_\beta'}\mathbb{E}_{\mathcal{D}}\log Z\bigg|_{\xi,\chi=0}, \quad\quad (\text{SI.1.50})$$

where the derivatives are with respect to $\xi_\alpha' = \xi_\alpha/\sqrt{\eta_\alpha}$ and $\chi_\alpha' = \chi_\alpha/\sqrt{\eta_\alpha}$, respectively. Taking derivatives for each entry of $\boldsymbol{\xi}'$, we obtain:

$$\frac{1}{\beta}\frac{\partial}{\partial\xi_\alpha'}\mathbb{E}_{\mathcal{D}}\log Z = \sqrt{\eta_\alpha}\bar{w}_\alpha - \frac{\kappa\sqrt{\eta_\alpha}(\bar{w}_\alpha - \xi_\alpha)}{P\eta_\alpha + \kappa} = \frac{P\eta_\alpha\sqrt{\eta_\alpha}\bar{w}_\alpha + \kappa\sqrt{\eta_\alpha}\xi_\alpha}{P\eta_\alpha + \kappa}. \quad\quad (\text{SI.1.51})$$

Hence the *average estimator* has the following form:

$$\mathbb{E}_{\mathcal{D}}f^*(\mathbf{x};P) = \sum_\rho \frac{P\eta_\rho}{P\eta_\rho + \kappa}\bar{w}_\rho\psi_\rho(\mathbf{x}), \quad\quad (\text{SI.1.52})$$

which approaches to the target function as $P\to\infty$. Note that the learned function can only express the components which span the RKHS. If the target function has out-of-RKHS components, those will never be learned.

Finally, we want to calculate the correlation function of the estimator. Given the partition function $Z = \int d\mathbf{w}e^{-\beta H(\mathbf{w};\mathcal{D})+\beta\boldsymbol{\xi}\cdot\mathbf{w}+\frac{\beta}{2}\boldsymbol{\chi}^\top\mathbf{w}\mathbf{w}^\top\boldsymbol{\chi}}$, notice that the variance:

$$\frac{1}{\beta^2}\frac{\partial^2}{\partial\xi_\alpha\partial\xi_\beta}\mathbb{E}_{\mathcal{D}}\log Z = \mathbb{E}_{\mathcal{D}}[\langle w_\alpha w_\beta\rangle_\mathbf{w} - \langle w_\alpha\rangle_\mathbf{w}\langle w_\beta\rangle_\mathbf{w}] = \frac{1}{\beta}\frac{\kappa}{P\eta_\alpha + \kappa}\delta_{\alpha\beta} \quad\quad (\text{SI.1.53})$$

vanishes as $\beta\to\infty$ since there is a unique solution. However, there is variance to the estimator due to averaging over different training sets which is given by:

$$\mathbb{E}_{\mathcal{D}}[\langle w_\alpha w_\beta\rangle_\mathbf{w}] - \mathbb{E}_{\mathcal{D}}\langle w_\alpha\rangle_\mathbf{w}\mathbb{E}_{\mathcal{D}}\langle w_\beta\rangle_\mathbf{w}, \quad\quad (\text{SI.1.54})$$

and it is finite as $\beta\to\infty$. The first term, the eigenfunction expansion coefficients of the correlation function of the estimator $\langle f(\mathbf{x})f(\mathbf{x}')\rangle$, can be calculated by taking two derivatives of $\mathbb{E}_{\mathcal{D}}\log Z$ with respect to $\boldsymbol{\chi}'$. To simplify the calculation, we first redefine $\mathbf{Q} \equiv \mathbf{I} + \frac{P}{\kappa}\boldsymbol{\Lambda} - \boldsymbol{\Lambda}^{1/2}\boldsymbol{\chi}'\boldsymbol{\chi}'^\top\boldsymbol{\Lambda}^{1/2}$ by setting $J = 0$ and introduce the notation $\partial_\alpha \equiv \frac{\partial}{\partial\chi_\alpha'}$ for notational simplicity. First, we calculate the derivatives of $\kappa$:

$$\partial_\alpha\kappa = -\operatorname{Tr}\boldsymbol{\Lambda}\mathbf{Q}^{-1}\left(-\frac{P}{\kappa^2}\boldsymbol{\Lambda}\partial_\alpha\kappa - \boldsymbol{\Lambda}^{1/2}\partial_\alpha(\boldsymbol{\chi}'\boldsymbol{\chi}'^\top)\boldsymbol{\Lambda}^{1/2}\right)\mathbf{Q}^{-1}$$

$$= \partial_\alpha\kappa\frac{P}{\kappa^2}\operatorname{Tr}\boldsymbol{\Lambda}^2\mathbf{Q}^{-2} + \operatorname{Tr}\boldsymbol{\Lambda}\mathbf{Q}^{-1}\boldsymbol{\Lambda}^{1/2}\partial_\alpha(\boldsymbol{\chi}'\boldsymbol{\chi}'^\top)\boldsymbol{\Lambda}^{1/2}\mathbf{Q}^{-1}$$

$$= \partial_\alpha\kappa\frac{P}{\kappa^2}\operatorname{Tr}\boldsymbol{\Lambda}^2\mathbf{Q}^{-2} + 2\sum_\rho\eta_\rho\mathbf{Q}_{\rho\alpha}^{-1}\sqrt{\eta_\alpha}\left(\sum_\sigma\sqrt{\eta_\sigma}\chi_\sigma'\mathbf{Q}_{\sigma\rho}^{-1}\right). \quad\quad (\text{SI.1.55})$$

Hence, we find that:

$$\partial_{\boldsymbol{\chi}'}\kappa|_{\boldsymbol{\chi}'=0} = \partial_{\boldsymbol{\chi}'}\mathbf{Q}|_{\boldsymbol{\chi}'=0} = 0. \tag{SI.1.56}$$

This greatly simplifies the second derivative of $\kappa$:

$$\partial_\alpha\partial_\beta\kappa|_{\boldsymbol{\chi}'=0} = \left[\partial_\alpha\partial_\beta\kappa\frac{P}{\kappa^2}\operatorname{Tr}\boldsymbol{\Lambda}^2\mathbf{Q}^{-2} + 2\sum_\rho \eta_\rho\mathbf{Q}_{\rho\alpha}^{-1}\sqrt{\eta_\alpha\eta_\beta}\mathbf{Q}_{\beta\rho}^{-1}\right]\Bigg|_{\boldsymbol{\chi}'=0}$$

$$= (\partial_\alpha\partial_\beta\kappa|_{\boldsymbol{\chi}'=0})\sum_\rho\frac{P\eta_\rho^2}{(P\eta_\rho+\kappa)^2} + \frac{2\kappa^2}{P}\frac{P\eta_\alpha^2}{(P\eta_\alpha+\kappa)^2}\delta_{\alpha\beta}$$

$$= \frac{2\kappa^2}{P}\frac{1}{1-\gamma}\frac{P\eta_\alpha^2}{(P\eta_\alpha+\kappa)^2}\delta_{\alpha\beta}, \tag{SI.1.57}$$

where $\gamma = \sum_\rho\frac{P\eta_\rho^2}{(P\eta_\rho+\kappa)^2}$ as defined before. Now we calculate the variance of the expected function:

$$\sqrt{\eta_\alpha\eta_\beta}\mathbb{E}_{\mathcal{D}}[w_\alpha^*(\mathcal{D})w_\beta^*(\mathcal{D})] = \lim_{\beta\to\infty}\frac{1}{\beta}\partial_\alpha\partial_\beta\mathbb{E}_{\mathcal{D}}\log Z\big|_{\boldsymbol{\xi}',\boldsymbol{\chi}'=0}$$

$$= \frac{P}{2}\frac{\varepsilon^2}{\kappa^2}\partial_\alpha\partial_\beta\kappa + \sqrt{\eta_\alpha\eta_\beta}\bar{w}_\alpha\bar{w}_\beta - \kappa\frac{\sqrt{\eta_\alpha\eta_\beta}\bar{w}_\alpha\bar{w}_\beta}{P\eta_\beta+\kappa} - \kappa\frac{\sqrt{\eta_\alpha\eta_\beta}\bar{w}_\alpha\bar{w}_\beta}{P\eta_\alpha+\kappa}$$

$$\quad - \frac{1}{2}\bar{\mathbf{W}}^\top\mathbf{G}^{-1}\left(-\frac{P}{\kappa^2}\boldsymbol{\Lambda}\partial_\alpha\partial_\beta\kappa - \boldsymbol{\Lambda}^{1/2}\partial_\alpha\partial_\beta(\boldsymbol{\chi}'\boldsymbol{\chi}'^\top)\boldsymbol{\Lambda}^{1/2}\right)\mathbf{G}^{-1}\bar{\mathbf{W}}$$

$$= \frac{1}{1-\gamma}\left(\varepsilon^2 + \kappa^2\sum_\rho\frac{\eta_\rho\bar{w}_\rho^2}{(P\eta_\rho+\kappa)^2}\right)\frac{P\eta_\alpha^2}{(P\eta_\alpha+\kappa)^2}\delta_{\alpha\beta} + \kappa^2\frac{\eta_\alpha\bar{w}_\alpha^2}{(P\eta_\alpha+\kappa)^2}\delta_{\alpha\beta}$$

$$\quad + \frac{P\eta_\beta\sqrt{\eta_\alpha\eta_\beta}\bar{w}_\alpha\bar{w}_\beta}{P\eta_\beta+\kappa} - \kappa\frac{\sqrt{\eta_\alpha\eta_\beta}\bar{w}_\alpha\bar{w}_\beta}{P\eta_\alpha+\kappa}. \tag{SI.1.58}$$

Now we can calculate the coefficients of covariance of the estimator:

$$\sqrt{\eta_\alpha\eta_\beta}\left(\mathbb{E}_{\mathcal{D}}[w_\alpha^*w_\beta^*] - \mathbb{E}_{\mathcal{D}}w_\alpha^*\mathbb{E}_{\mathcal{D}}w_\beta^*\right) \tag{SI.1.59}$$

$$= \frac{1}{1-\gamma}\left(\varepsilon^2 + \kappa^2\sum_\rho\frac{\eta_\rho\bar{w}_\rho^2}{(P\eta_\rho+\kappa)^2}\right)\frac{P\eta_\alpha^2}{(P\eta_\alpha+\kappa)^2}\delta_{\alpha\beta} + \kappa^2\frac{\eta_\alpha\bar{w}_\alpha^2}{(P\eta_\alpha+\kappa)^2}\delta_{\alpha\beta}$$

$$\quad + \frac{(P\eta_\alpha+\kappa)P\eta_\beta\sqrt{\eta_\alpha\eta_\beta}\bar{w}_\alpha\bar{w}_\beta}{(P\eta_\alpha+\kappa)(P\eta_\beta+\kappa)} - \kappa\frac{(P\eta_\beta+\kappa)\sqrt{\eta_\alpha\eta_\beta}\bar{w}_\alpha\bar{w}_\beta}{(P\eta_\alpha+\kappa)(P\eta_\beta+\kappa)} - \frac{P^2\eta_\alpha\eta_\beta\sqrt{\eta_\alpha\eta_\beta}\bar{w}_\alpha\bar{w}_\beta}{(P\eta_\alpha+\kappa)(P\eta_\beta+\kappa)}$$

$$= \boxed{\frac{1}{1-\gamma}\left(\varepsilon^2 + \kappa^2\sum_\rho\frac{\eta_\rho\bar{w}_\rho^2}{(P\eta_\rho+\kappa)^2}\right)\frac{P\eta_\alpha^2}{(P\eta_\alpha+\kappa)^2}\delta_{\alpha\beta} + \kappa^2\frac{\sqrt{\eta_\alpha\eta_\beta}\bar{w}_\alpha\bar{w}_\beta}{(P\eta_\alpha+\kappa)(P\eta_\beta+\kappa)}(\delta_{\alpha\beta}-1)}.$$

Hence the covariance of the estimator is:

$$Cov[\langle f^*(\mathbf{x};P)f^*(\mathbf{x}';P)\rangle_{\mathcal{D}}] = \sum_{\alpha\beta}\left(\mathbb{E}_{\mathcal{D}}[w_\alpha^*w_\beta^*] - \mathbb{E}_{\mathcal{D}}w_\alpha^*\mathbb{E}_{\mathcal{D}}w_\beta^*\right)\psi_\alpha(\mathbf{x})\psi_\beta(\mathbf{x}'). \tag{SI.1.60}$$

## SI.1.5 Generalization Error

Having computed the mean and covariance of the estimator, now we can calculate the average generalization error which can be decomposed as:

$$\mathbb{E}_{\mathcal{D}} E_g = \int d\mathbf{x}\, \tilde{p}(\mathbf{x}) \mathbb{E}_{\mathcal{D}}(f^{*2}(\mathbf{x})) - 2 \int d\mathbf{x}\, \tilde{p}(\mathbf{x}) \mathbb{E}_{\mathcal{D}} f^*(\mathbf{x}) \bar{f}(\mathbf{x}) + \int d\mathbf{x}\, \tilde{p}(\mathbf{x}) \bar{f}(\mathbf{x})^2, \quad \text{(SI.1.61)}$$

where we compute the data average over a new distribution $\tilde{p}(\mathbf{x})$. A useful quantity is the overlap matrix defined as:

$$\mathscr{O}_{\rho\gamma} = \int d\mathbf{x}\, \tilde{p}(\mathbf{x}) \phi_\rho(\mathbf{x}) \phi_\gamma(\mathbf{x}), \quad \text{(SI.1.62)}$$

and $\gamma' = \sum_\rho \mathscr{O}_{\rho\rho} \frac{P\eta_\rho^2}{(P\eta_\rho + \kappa)^2}$. In terms of these quantities, using the calculation above, we find:

$$\int d\mathbf{x}\, \tilde{p}(\mathbf{x}) \mathbb{E}_{\mathcal{D}}(f^{*2}(\mathbf{x})) - \int d\mathbf{x}\, \tilde{p}(\mathbf{x}) \mathbb{E}_{\mathcal{D}} f^*(\mathbf{x}) \mathbb{E}_{\mathcal{D}} f^*(\mathbf{x}) = \frac{\gamma'}{1-\gamma}\left(\varepsilon^2 + \kappa^2 \sum_\rho \frac{\eta_\rho \bar{w}_\rho^2}{(P\eta_\rho + \kappa)^2}\right)$$

$$\int d\mathbf{x}\, \tilde{p}(\mathbf{x}) \mathbb{E}_{\mathcal{D}} f^*(\mathbf{x}) \mathbb{E}_{\mathcal{D}} f^*(\mathbf{x}) = \sum_{\rho,\gamma} \mathscr{O}_{\rho\gamma} \frac{P\eta_\rho^{3/2} \bar{w}_\rho}{P\eta_\rho + \kappa} \frac{P\eta_\gamma^{3/2} \bar{w}_\gamma}{P\eta_\gamma + \kappa}$$

$$\int d\mathbf{x}\, \tilde{p}(\mathbf{x}) \mathbb{E}_{\mathcal{D}} f^*(\mathbf{x}) \bar{f}(\mathbf{x}) = \sum_{\rho,\gamma} \mathscr{O}_{\rho\gamma} \sqrt{\eta_\rho} \bar{w}_\rho \frac{P\eta_\gamma^{3/2} \bar{w}_\gamma}{P\eta_\gamma + \kappa}$$

$$\int d\mathbf{x}\, \tilde{p}(\mathbf{x}) \bar{f}(\mathbf{x})^2 = \sum_{\rho,\gamma} \mathscr{O}_{\rho\gamma} \sqrt{\eta_\rho} \bar{w}_\rho \sqrt{\eta_\gamma} \bar{w}_\gamma, \quad \text{(SI.1.63)}$$

where the first line is the contribution to generalization error due to the estimator variance. Hence generalization error is:

$$\mathbb{E}_{\mathcal{D}} E_g = \underbrace{\frac{\gamma'}{1-\gamma}\left(\varepsilon^2 + \kappa^2 \sum_\rho \frac{\bar{a}_\rho^2}{(P\eta_\rho + \kappa)^2}\right)}_{\text{Variance } V} + \underbrace{\kappa^2 \sum_{\rho\gamma} \mathscr{O}_{\rho\gamma} \frac{\bar{a}_\rho}{P\eta_\rho + \kappa} \frac{\bar{a}_\gamma}{P\eta_\gamma + \kappa}}_{\text{Bias } B} \quad \text{(SI.1.64)}$$

where we replaced $\bar{a}_\rho = \sqrt{\eta_\rho} \bar{w}_\rho$ which are the $L^2$ weights of the target. This is the bias-variance decomposition of generalization error in our setting where the bias term is monotonically decreasing while the variance term is solely responsible for any non-monotonicity appearing in the generalization error.

This expression also shows simply how to handle out-of-RKHS components of the target function. Assume that the kernel is band-limited, meaning $\eta_{\rho>N} = 0$ for some $N < M$. Then generalization error becomes:

$$\mathbb{E}_{\mathcal{D}} E_g = \frac{\gamma'}{1-\gamma}\left(\tilde{\varepsilon}^2 + \kappa^2 \sum_{\rho=1}^{N} \frac{\bar{a}_\rho^2}{(P\eta_\rho + \kappa)^2}\right) + \kappa^2 \sum_{\rho,\gamma=1}^{N} \mathscr{O}_{\rho\gamma} \frac{\bar{a}_\rho}{P\eta_\rho + \kappa} \frac{\bar{a}_\gamma}{P\eta_\gamma + \kappa}$$

$$+ 2\kappa \sum_{\rho=N+1}^{M} \sum_{\gamma=1}^{N} \mathscr{O}_{\rho\gamma} \frac{\bar{a}_\rho \bar{a}_\gamma}{P\eta_\gamma + \kappa} + \sum_{\rho,\gamma=N+1}^{M} \mathscr{O}_{\rho\gamma} \bar{a}_\rho \bar{a}_\gamma, \quad \text{(SI.1.65)}$$

where we define the effective noise $\tilde{\varepsilon}^2 = \varepsilon^2 + \sum_{\rho=N+1}^{M} \bar{a}_\rho$. Therefore, bias decomposes into three terms which correspond to three block components of the overlap matrix. The last diagonal block of the overlap matrix yields an irreducible error on the generalization error.

We are mostly interested in how out-of-distribution generalization deviates from when the training and test distributions are same. The in-distribution generalization is simply given by setting overlap matrix to identity:

$$E_g^{0,p(\mathbf{x})} = \frac{\gamma}{1-\gamma}\left(\varepsilon^2 + \kappa^2 \sum_{\rho=1}^{N} \frac{\bar{a}_\rho^2}{(P\eta_\rho + \kappa)^2}\right) + \kappa^2 \sum_{\rho=1}^{N} \frac{\bar{a}_\rho^2}{(P\eta_\rho + \kappa)^2}, \quad \text{(SI.1.66)}$$

where $E_g^{0,p(\mathbf{x})}$ denotes the generalization error when both training and test distributions are $p(\mathbf{x})$. Note that the data $\{\bar{a}_\rho\}$ and $\{\eta_\rho\}$ are obtained with respect to $p(\mathbf{x})$ and can be replaced by $\{\tilde{\bar{a}}_\rho\}$ and $\{\tilde{\eta}_\rho\}$ if the test distribution is fixed and the training distribution is varied. We will first consider the former case with fixed training distribution and varying test distribution:

$$\Delta E_g \equiv E_g - E_g^{0,p(\mathbf{x})} = \frac{\gamma' - \gamma}{1 - \gamma}\varepsilon^2 + \kappa^2\bar{\mathbf{a}}^\top(P\mathbf{\Lambda} + \kappa\mathbf{I})^{-1}\mathscr{O}'(P\mathbf{\Lambda} + \kappa\mathbf{I})^{-1}\bar{\mathbf{a}} \qquad \text{(SI.1.67)}$$

where we defined $\mathscr{O}' = \mathscr{O} - \frac{1-\gamma'}{1-\gamma}\mathbf{I}$ which captures the effect of distribution mismatch on the generalization error. An example of shifted overlap matrix $\mathscr{O}'$ has been shown in Figure 1 and Figure 2.

### SI.1.6 Symmetries of Overlap Matrix

The overlap matrix naturally arises when one considers the kernel eigenvalue problem with respect to two different input distributions:

$$\int d\mathbf{x}' \, p(\mathbf{x}')K(\mathbf{x},\mathbf{x}')\phi_\rho(\mathbf{x}') = \eta_\rho\phi_\rho(\mathbf{x}), \quad \Rightarrow \quad K(\mathbf{x},\mathbf{x}') = \sum_\rho \eta_\rho\phi_\rho(\mathbf{x})\phi_\rho(\mathbf{x})$$

$$\int d\mathbf{x}' \, \tilde{p}(\mathbf{x}')K(\mathbf{x},\mathbf{x}')\tilde{\phi}_\rho(\mathbf{x}') = \tilde{\eta}_\rho\tilde{\phi}_\rho(\mathbf{x}), \quad \Rightarrow \quad K(\mathbf{x},\mathbf{x}') = \sum_\rho \tilde{\eta}_\rho\tilde{\phi}_\rho(\mathbf{x})\tilde{\phi}_\rho(\mathbf{x}) \qquad \text{(SI.1.68)}$$

Therefore we have two sets of orthonormal bases $\{\phi_\rho\}$ and $\{\tilde{\phi}_\rho\}$ with respect to distributions $p(\mathbf{x})$ and $\tilde{p}(\mathbf{x})$, both spanning $L^2(\mathcal{X})$. A square integrable function in this space can be expanded as:

$$f(\mathbf{x}) = \sum_\rho a_\rho\phi_\rho(\mathbf{x}) = \sum_\rho \tilde{a}_\rho\tilde{\phi}_\rho(\mathbf{x}), \quad \langle\phi_\rho,\phi_\gamma\rangle_p = \delta_{\rho\gamma}, \quad \langle\tilde{\phi}_\rho,\tilde{\phi}_\gamma\rangle_{\tilde{p}} = \delta_{\rho\gamma}, \qquad \text{(SI.1.69)}$$

where we defined the inner product $\langle f,g\rangle_{p(\mathbf{x})} = \int d\mathbf{x}\, p(\mathbf{x})f(\mathbf{x})g(\mathbf{x})$. Expansion coefficients can be found in terms of each other via:

$$\tilde{\mathbf{a}}_\rho = \sum_\gamma a_\gamma \langle\phi_\gamma,\tilde{\phi}_\rho\rangle_{\tilde{p}} = (\tilde{\mathbf{A}}\mathbf{a})_\rho, \quad \tilde{\mathbf{A}}_{\rho\gamma} \equiv \langle\tilde{\phi}_\rho,\phi_\gamma\rangle_{\tilde{p}},$$

$$\mathbf{a}_\rho = \sum_\gamma \tilde{a}_\gamma \langle\tilde{\phi}_\gamma,\phi_\rho\rangle_p = (\mathbf{A}\tilde{\mathbf{a}})_\rho, \quad \mathbf{A}_{\rho\gamma} \equiv \langle\phi_\rho,\tilde{\phi}_\gamma\rangle_p. \qquad \text{(SI.1.70)}$$

This immediately implies that:

$$\tilde{\mathbf{A}}\mathbf{A} = \mathbf{A}\tilde{\mathbf{A}} = \mathbf{I} \qquad \text{(SI.1.71)}$$

We call $\mathbf{A}, \tilde{\mathbf{A}}$ cross-overlap matrices and in general they do not correspond to norm-preserving transformations. Note that the $L^2$ norm of a function depends on the probability measure on the space:

$$\|f\|_{p(\mathbf{x})}^2 = \sum_\rho \mathbf{a}_\rho^2 \neq \sum_\rho (\tilde{\mathbf{A}}\mathbf{a})_\rho^2 = \|f\|_{\tilde{p}(\mathbf{x})}^2 \qquad \text{(SI.1.72)}$$

Later we show that the Hilbert norm $\|f\|_{\mathcal{H}}$ is independent of the probability measures if $f \in \mathcal{H}$.

These matrices also connect the eigenfunctions:

$$\mathbf{\Phi} = \tilde{\mathbf{A}}^\top\tilde{\mathbf{\Phi}}, \quad \tilde{\mathbf{\Phi}} = \mathbf{A}^\top\mathbf{\Phi} \qquad \text{(SI.1.73)}$$

Using these relations, cross-overlap matrices can be related to overlap matrix $\mathscr{O}$:

$$\mathscr{O}_{\rho\gamma} = \int d\mathbf{x}\,\tilde{p}(\mathbf{x})\phi_\rho(\mathbf{x})\phi_\gamma(\mathbf{x}) = \sum_{\gamma'} \tilde{\mathbf{A}}_{\gamma\gamma'}^\top \int d\mathbf{x}\,\tilde{p}(\mathbf{x})\phi_\rho(\mathbf{x})\tilde{\phi}_{\gamma'}(\mathbf{x}) = (\tilde{\mathbf{A}}^\top\tilde{\mathbf{A}})_{\rho\gamma}$$

$$\tilde{\mathscr{O}}_{\rho\gamma} = \int d\mathbf{x}\,p(\mathbf{x})\tilde{\phi}_\rho(\mathbf{x})\tilde{\phi}_\gamma(\mathbf{x}) = \sum_{\gamma'} \mathbf{A}_{\gamma\gamma'}^\top \int d\mathbf{x}\,p(\mathbf{x})\tilde{\phi}_\rho(\mathbf{x})\phi_{\gamma'}(\mathbf{x}) = (\mathbf{A}^\top\mathbf{A})_{\rho\gamma}, \qquad \text{(SI.1.74)}$$

which, together with Eq.(SI.1.71), have inverses $\mathscr{O}^{-1} = \mathbf{A}\mathbf{A}^\top$ and $\tilde{\mathscr{O}}^{-1} = \tilde{\mathbf{A}}\tilde{\mathbf{A}}^\top$. Furthermore, this shows that $\mathscr{O}$ and $\tilde{\mathscr{O}}$ are symmetric positive definite matrices.

Now we connect these matrices to the kernel eigenvalues. Kernel features defined by $\boldsymbol{\Psi}(\mathbf{x}) = \boldsymbol{\Lambda}^{1/2}\boldsymbol{\Phi}(\mathbf{x})$ are orhonormal with respect to the Hilbert inner product on the RKHS but their $L^2$ inner product depends on the probability measure:

$$\langle \boldsymbol{\Psi}(\mathbf{x}), \boldsymbol{\Phi}(\mathbf{x})^\top \rangle_{\mathcal{H}} = \mathbf{I}, \quad \int d\mathbf{x}\, p(\mathbf{x})\boldsymbol{\Psi}(\mathbf{x})\boldsymbol{\Psi}(\mathbf{x})^\top = \boldsymbol{\Lambda}$$

$$\langle \tilde{\boldsymbol{\Psi}}(\mathbf{x}), \tilde{\boldsymbol{\Phi}}(\mathbf{x})^\top \rangle_{\mathcal{H}} = \mathbf{I}, \quad \int d\mathbf{x}\, \tilde{p}(\mathbf{x})\tilde{\boldsymbol{\Psi}}(\mathbf{x})\tilde{\boldsymbol{\Psi}}(\mathbf{x})^\top = \tilde{\boldsymbol{\Lambda}}, \tag{SI.1.75}$$

where $\langle .,. \rangle_{\mathcal{H}}$ is defined in Eq.(SI.1.5). In terms of features, kernel can be expressed as:

$$K(\mathbf{x}, \mathbf{x}') = \boldsymbol{\Psi}(\mathbf{x})^\top \boldsymbol{\Psi}(\mathbf{x}') = \tilde{\boldsymbol{\Psi}}(\mathbf{x})^\top \tilde{\boldsymbol{\Psi}}(\mathbf{x}'). \tag{SI.1.76}$$

We note that the Hilbert inner product of two functions $f, g \in \mathcal{H}$ does not depend on the measure against which the kernel is diagonalized [26]. Hence for any function $f(\mathbf{x}) = \mathbf{w}^\top \boldsymbol{\Psi}(\mathbf{x}) = \tilde{\mathbf{w}}^\top \tilde{\boldsymbol{\Psi}}(\mathbf{x})$ we have:

$$\|f\|_{\mathcal{H}}^2 = \mathbf{w}^\top \mathbf{w} = \tilde{\mathbf{w}}^\top \tilde{\mathbf{w}}, \tag{SI.1.77}$$

which immediately implies that there exists an orthogonal transformation $U$ which rotates the features and the weights as follows:

$$\tilde{\boldsymbol{\Psi}}(\mathbf{x}) = U\boldsymbol{\Psi}(\mathbf{x}), \quad \tilde{\mathbf{w}} = U\mathbf{w}, \tag{SI.1.78}$$

where $U^\top = U^{-1}$. Using the relations in Eq.(SI.1.75), one can obtain the relations:

$$\boldsymbol{\Lambda}^{1/2}\mathscr{O}\boldsymbol{\Lambda}^{1/2} = U^\top \tilde{\boldsymbol{\Lambda}} U$$

$$\tilde{\boldsymbol{\Lambda}}^{1/2}\tilde{\mathscr{O}}\tilde{\boldsymbol{\Lambda}}^{1/2} = U\boldsymbol{\Lambda} U^\top. \tag{SI.1.79}$$

Furthermore, one can easily find $U$ in terms of matrices $\mathbf{A}, \tilde{\mathbf{A}}$:

$$U = \boldsymbol{\Lambda}^{1/2}\tilde{\mathbf{A}}^\top \tilde{\boldsymbol{\Lambda}}^{-1/2}, \quad U^\top = \tilde{\boldsymbol{\Lambda}}^{1/2}\mathbf{A}^\top \boldsymbol{\Lambda}^{-1/2} \tag{SI.1.80}$$

Of course this relation requires eigenvalue matrices to be invertible otherwise one must replace inverses with pseudo-inverses.

Finally, we obtain how the two eigenvalue matrices connect to each other:

$$K(\mathbf{x}, \mathbf{x}') = \boldsymbol{\Phi}^\top \boldsymbol{\Lambda} \boldsymbol{\Phi} = \tilde{\boldsymbol{\Phi}}^\top \tilde{\boldsymbol{\Lambda}} \tilde{\boldsymbol{\Phi}} \;\Rightarrow\; \tilde{\boldsymbol{\Lambda}} = \tilde{\mathbf{A}}\boldsymbol{\Lambda}\tilde{\mathbf{A}}^\top, \quad \boldsymbol{\Lambda} = \mathbf{A}\tilde{\boldsymbol{\Lambda}}\mathbf{A}^\top. \tag{SI.1.81}$$

These transformations help us to rewrite the generalization error in terms of the test eigenvalues and weights.

### SI.1.7  Generalization Error In Terms of Test Distribution

Using the identities in SI.1.6, we first start with expressing $\kappa$ in terms of the test distribution eigenvalues:

$$\kappa = \lambda + \kappa \operatorname{Tr}\left\{ \boldsymbol{\Lambda}(P\boldsymbol{\Lambda} + \kappa\mathbf{I})^{-1} \right\} = \lambda + \kappa \operatorname{Tr}\left\{ \tilde{\boldsymbol{\Lambda}}(P\tilde{\boldsymbol{\Lambda}} + \kappa\tilde{\mathscr{O}}^{-1})^{-1} \right\}, \tag{SI.1.82}$$

where we used the identity $\tilde{\boldsymbol{\Lambda}}^{1/2}\tilde{\mathscr{O}}\tilde{\boldsymbol{\Lambda}}^{1/2} = U\boldsymbol{\Lambda} U^\top$ and the properties of trace. Similarly we have:

$$\gamma = P\operatorname{Tr}\left( \boldsymbol{\Lambda}^2(P\boldsymbol{\Lambda} + \kappa\mathbf{I})^{-2} \right) = P\operatorname{Tr}\left( \tilde{\boldsymbol{\Lambda}}(P\tilde{\boldsymbol{\Lambda}} + \kappa\tilde{\mathscr{O}}^{-1})^{-1}\tilde{\boldsymbol{\Lambda}}(P\tilde{\boldsymbol{\Lambda}} + \kappa\tilde{\mathscr{O}}^{-1})^{-1} \right)$$

$$\gamma' = P\operatorname{Tr}\left( \mathscr{O}\boldsymbol{\Lambda}^2(P\boldsymbol{\Lambda} + \kappa\mathbf{I})^{-2} \right) = P\operatorname{Tr}\left( \mathscr{O}\tilde{\boldsymbol{\Lambda}}(P\tilde{\boldsymbol{\Lambda}} + \kappa\tilde{\mathscr{O}}^{-1})^{-1}\tilde{\boldsymbol{\Lambda}}(P\tilde{\boldsymbol{\Lambda}} + \kappa\tilde{\mathscr{O}}^{-1})^{-1} \right) \tag{SI.1.83}$$

Generalization error is given by:

$$E_g = \frac{\gamma'}{1 - \gamma}\left( \varepsilon^2 + \kappa^2 \bar{\mathbf{a}}^\top (P\boldsymbol{\Lambda} + \kappa\mathbf{I})^{-2}\bar{\mathbf{a}} \right) + \kappa^2 \bar{\mathbf{a}}^\top (P\boldsymbol{\Lambda} + \kappa\mathbf{I})^{-1}\mathscr{O}(P\boldsymbol{\Lambda} + \kappa\mathbf{I})^{-1}\bar{\mathbf{a}}. \tag{SI.1.84}$$

Using the relations $\mathbf{\Lambda} = \mathbf{A}\tilde{\mathbf{\Lambda}}\mathbf{A}^\top$, $\tilde{\mathbf{a}} = \tilde{\mathbf{A}}\mathbf{a}$ and $\tilde{\mathbf{A}}\tilde{\mathbf{A}}^\top = \tilde{\mathscr{O}}^{-1}$, we get:

$$E_g = \frac{\gamma'}{1-\gamma}\big(\varepsilon^2 + \kappa^2\tilde{\mathbf{a}}^\top(P\tilde{\mathbf{\Lambda}} + \kappa\tilde{\mathscr{O}}^{-1})^{-1}\tilde{\mathscr{O}}^{-1}(P\tilde{\mathbf{\Lambda}} + \kappa\tilde{\mathscr{O}}^{-1})^{-1}\tilde{\mathbf{a}}\big)$$
$$+ \kappa^2\tilde{\mathbf{a}}^\top(P\tilde{\mathbf{\Lambda}} + \kappa\tilde{\mathscr{O}}^{-1})^{-1}\tilde{\mathscr{O}}^{-2}(P\tilde{\mathbf{\Lambda}} + \kappa\tilde{\mathscr{O}}^{-1})^{-1}\tilde{\mathbf{a}}. \tag{SI.1.85}$$

As we have done before, we can compare this to in-distribution generalization error when both training and test distributions are $\tilde{p}(\mathbf{x})$:

$$E_g^{0,\tilde{p}(\mathbf{x})} = \frac{\tilde{\gamma}}{1-\tilde{\gamma}}\left(\varepsilon^2 + \tilde{\kappa}^2\sum_{\rho=1}^{N}\frac{\tilde{\tilde{a}}_\rho^2}{(P\tilde{\eta}_\rho + \tilde{\kappa})^2}\right) + \tilde{\kappa}^2\sum_{\rho=1}^{N}\frac{\tilde{\tilde{a}}_\rho^2}{(P\tilde{\eta}_\rho + \tilde{\kappa})^2}, \tag{SI.1.86}$$

where $\tilde{\kappa} = \lambda + \tilde{\kappa}\,\mathrm{Tr}\big(\tilde{\mathbf{\Lambda}}(P\tilde{\mathbf{\Lambda}} + \tilde{\kappa}\mathbf{I})^{-1}\big)$ and $\tilde{\gamma} = P\,\mathrm{Tr}\big(\tilde{\mathbf{\Lambda}}^2(P\tilde{\mathbf{\Lambda}} + \tilde{\kappa}\mathbf{I})^{-2}\big)$.

## SI.2 Further Analysis for Real Data Applications and Additional Experiments

The form for OOD generalization with label noise $\varepsilon^2 = 0$ given by Eq.(1) has a simple form in terms of the overlap matrix:

$$\Delta E_g = \mathbf{v}(P)^\top \mathscr{O}'\mathbf{v}(P)$$
$$\mathscr{O}' = \mathscr{O} - \mathbf{I} + \frac{\mathrm{Tr}\,(\mathscr{O} - \mathbf{I})\mathbf{M}(P)}{1 - \mathrm{Tr}\,\mathbf{M}(P)}\mathbf{I}$$
$$\mathscr{O}_{\rho\gamma} = \int d\mathbf{x}\,\tilde{p}(\mathbf{x})\phi_\rho(\mathbf{x})\phi_\gamma(\mathbf{x}), \tag{SI.2.1}$$

where $\mathbf{v}$ and $\mathbf{M}$ are independent of the test distribution and only given by target weights and kernel eigenvalues with respect to training distribution. In this section, we further study how mismatched train and test distributions can improve generalization.

**Fixed Training Distribution:** Consider minimizing $\Delta E_g$ in Eq.(SI.2.1) with respect to the overlap matrix. For fixed training distribution, the gradient with respect to $\mathscr{O}$ will be always positive-definite since the overlap matrix appears linearly in $\Delta E_g$ and is the only test distribution related quantity. This implies that $\Delta E_g$ does not have a non-trivial extremum except for the trivial $\Delta E_g = -E_g^{0,p(\mathbf{x})}$ which is possible only if $\mathscr{O} = 0$. This implies that OOD generalization can be reduced to zero and often never happens except for very special cases where all eigenfunctions become zero at some point $\mathbf{x}^*$ ($\phi_\rho(\mathbf{x}^*) = 0$) and the test distribution sharply centers around the same point $\mathbf{x}^*$ ($\tilde{p}(\mathbf{x}) = \delta(\mathbf{x} - \mathbf{x}^*)$).

However, a more likely scenario where $\Delta E_g$ is minimized is picking a test distribution which sharply centers around a point $\mathbf{x}^*$ which yields the smallest error:

$$\mathbf{x}^* = \min_{x \in \mathcal{D}}(f^*(\mathbf{x}) - \bar{f}(\mathbf{x}))^2. \tag{SI.2.2}$$

Then keep testing on this single point (in the sense that $\tilde{p}(\mathbf{x}) = \delta(\mathbf{x} - \mathbf{x}^*)$) should yield the best OOD generalization hence minimizing $\Delta E_g$. This has been noted in [4] and also what we observe in our experiments when we run gradient descent with respect to test distribution on $E_g$ for fixed training distribution. In Figure SI.2.1, we show an experiment where we fix the training distribution to be uniform and obtain beneficial/detrimental test distributions by running gradient descent/ascent on $E_g$ for different epochs. In Figure SI.2.1 (a), we see that when number of epochs is small, the test distribution stays close to the uniform distribution. But with increasing epochs, it sharpens more and more around a particular point. Here we also show that images which have high probability mass in beneficial distribution have low probability mass in detrimental distribution. In Figure SI.2.1 (b,c), we show that the empirical generalization error gets smaller/larger for beneficial/detrimental test distributions and matches perfectly with our theory.

**Fixed Test Distribution:** The other way $\Delta E_g$ can be minimized is by fixing test distribution and varying training distribution. Note that, for fixed test distribution $\tilde{p}(\mathbf{x})$, $\nabla_{\mathscr{O}|\tilde{p}(\mathbf{x})}\Delta E_g$ is much harder to evaluate since the only way we can change overlap matrix is by changing the training distribution which in turn alters all training distribution related quantities. Hence, a priori, it is not obvious if the

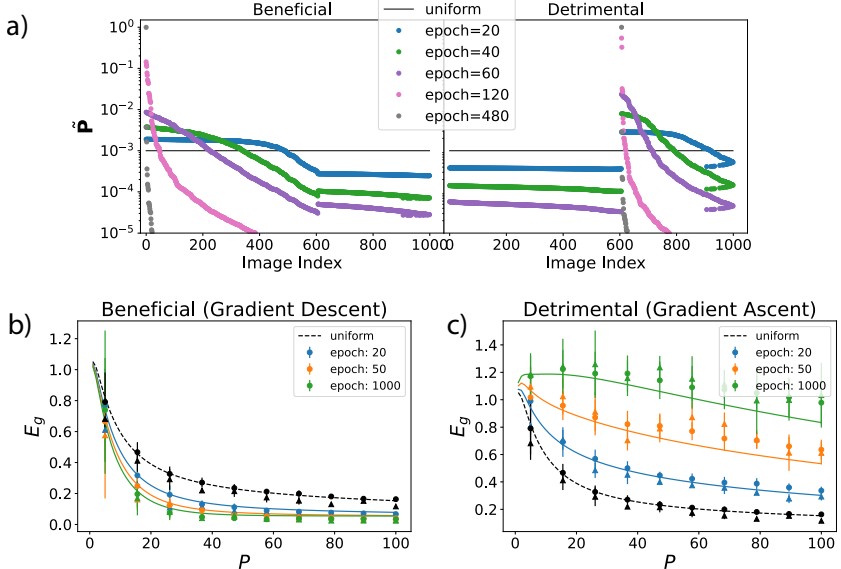

Figure SI.2.1: Optimal test distributions for fixed uniform training distribution. a) As the number of epochs increase for gradient descent/ascent, the test distribution concentrates more and more around a particular sample. We also see that the images which have high probability mass in beneficial test distribution are assigned low probability mass during gradient ascent. b) The generalization error curves for a trained neural network and kernel regression with the corresponding kernel. Both experiments match perfectly with theory (dashed lines). We see that beneficial test distributions yield better generalization than the in-distribution generalization. c) Same experiment with detrimental test distributions. As expected the generalization is worse than in-distribution case. The errorbars show standard deviation for 50 trials.

gradient descent procedure explained in Algorithm 1 converges to a fixed training distribution. Next, we show that it indeed converges to a fixed training measure for MNIST experiments with NTK and we discuss its implications.

We first note that our formula for generalization error depends on the number of training set size $P$ and hence the optimized training/test measures. Then our problem is what is the optimal choice of training examples for a limited training budget $P_{\text{budget}}$ so that generalization performance is maximal. This problem intuitively makes sense; for kernel ridgeless regression it would be the best to train uniformly on all samples if we had infinite training budget since we can exactly fit all samples. Or conversely, for few shot learning (low $P_{\text{budget}}$), we would like to choose the best few training samples which yields the best generalization performance.

In Figure SI.2.2, we show experiments where the test distributions is fixed to uniform and optimal training distributions are obtained via gradient descent on $\Delta \boldsymbol{z} \propto -\nabla_{\mathscr{O}|\tilde{p}(\mathbf{x})}\Delta E_g(P_{\text{budget}})$ for several training budgets where $\boldsymbol{z}$ are logits which translate to training probability masses via $\boldsymbol{p} = \text{softmax}(\boldsymbol{z})$. In Figure SI.2.2(a), we show that in fact the gradient descent converges to a fixed training distribution for all $P_{\text{budget}}$'s. Furthermore, we compute the participation ratio given by $\frac{1}{\sum_{\mu}(p^{\mu})^2}$ which quantifies how uniform the probability masses are. We find that this quantity approaches roughly to $P_{\text{budget}}$ meaning that GD finds $\sim P_{\text{budget}}$ samples which are most beneficial for generalization and discards the rest. In Figure SI.2.2(b), we show the training distributions for each $P_{\text{budget}}$ where each distribution is sorted from high to low probability masses. We again see that only $\sim P_{\text{budget}}$ examples have high probability mass and the rest are effectively ignored. The resulting high and low probability mass images are shown in Figure SI.2.2(c). We find that for low $P_{\text{budget}}$, easier samples are preferred to train on but for high $P_{\text{budget}}$'s we do not see an apparent qualitative difference between low and high probability mass images. Finally in Figure SI.2.2(d), we test if these optimized training distributions help generalization. We find that the training distributions optimized for a certain $P_{\text{budget}}$ performs

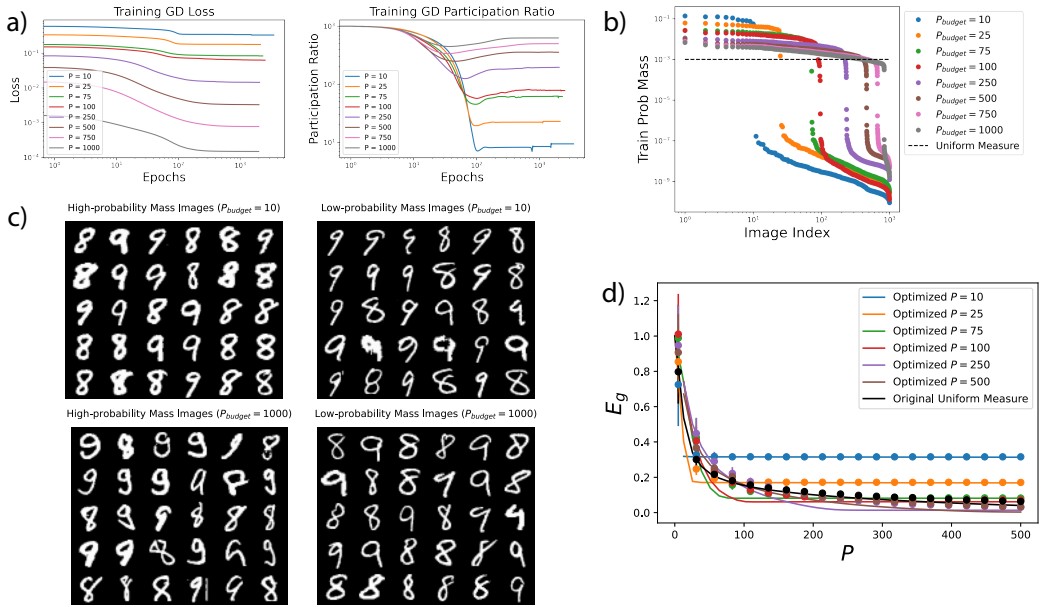

Figure SI.2.2: Optimal training distributions for fixed test distribution. a) Gradient descent on $E_g(P)$ with respect to training distribution converges for various $P_{\text{budget}}$. We also see that the participation ratio converges to $\sim P_{\text{budget}}$, implying that gradient descent finds the best $P_{\text{budget}}$ samples for fixed training budget to obtain an optimal generalization error. b) The training distributions for several $P_{\text{budget}}$'s. For larger training budgets, the training distribution approaches to the test distribution which is uniform. c) High and low probability mass images for $P_{\text{budget}} = \{10, 1000\}$. For smaller budgets ($P = 10$), GD finds the simplest examples to train on. For high training budgets ($P = 1000$), there is no qualitative difference. d) The corresponding generalization error for each optimized training distribution compared to kernel regression experiments on NTK (dots).

the best until the number of training samples hit $P_{\text{budget}}$ after which the generalization error stays constant.

## SI.2.1 Adversarial Attacks during Testing

We finally consider how our theory can be used in more practical settings such as adversarial attacks during testing. We devise a simple experiment where a kernel machine is trained on a subset of MNIST dataset with uniform training distribution and tested on the same subset but with noise added on the images. Then we identify the images which were correctly classified before adding noise but misclassified after. We create a new dataset where we add these misclassified images to the original dataset and run gradient descent/ascent on the generalization error to see if these images will be assigned low/high probability masses. We fix the training distribution to be uniform on the original images and have zero probability mass on the adversarial samples. On the other hand, the test probability masses stay as variables for the adversarial samples as well as the original samples.

In Figure SI.2.3 (a), we show the probability masses of each adversarial MNIST images obtained after gradient descent/ascent. We find that the beneficial test distribution places considerably low probability mass to the adversarial images than the detrimental test distribution. Figure SI.2.3 (b) shows the first few high and low probability mass images during gradient descent, where the adversarial examples are among the low probability mass images.

## SI.3 Linear Regression

As we discussed in the main text, it is straightforward to show that the generalization error reduces to the Eq.(4) when the input distributions are Gaussian with arbitrary covariance matrices $\mathbf{C}$ and $\tilde{\mathbf{C}}$ for training and test distributions.

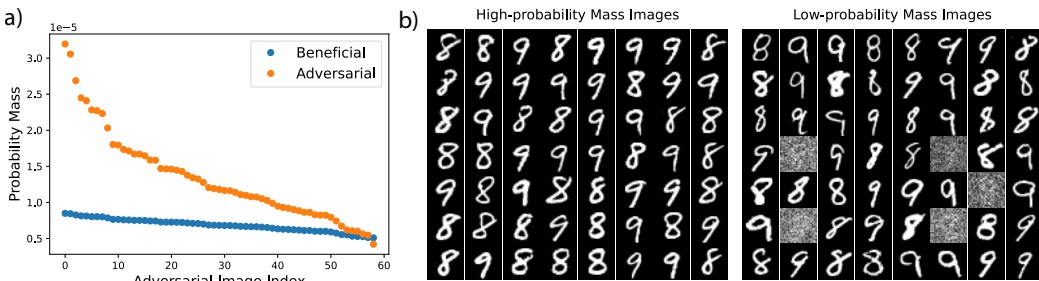

Figure SI.2.3: Adversarial samples in optimal test distribution. a) Probability masses of the adversarial samples after running gradient descent/ascent with respect to test distribution. Gradient descent/ascent places low/high probability mass to adversarial examples. b) The high and low probability mass images obtained after gradient descent. The adversarial examples get low probability mass.

Here, we generalize the discussion of linear regression with diagonal covariance matrices to include the out-of-RKHS scenarios. Similarly, we consider $D$-dimensional inputs and a linear target of the form $\bar{f} = \sum_{\rho=1}^{D} \beta_\rho x_\rho$. Furthermore, we take the training and test distributions to be of the form:

$$\mathbf{C} = \text{diag}\,(\underbrace{\sigma^2, \dots \sigma^2}_{M_r}, \underbrace{0, \dots 0}_{D-M_r})$$

$$\tilde{\mathbf{C}} = \text{diag}\,(\underbrace{\tilde{\sigma}^2, \dots \tilde{\sigma}^2}_{M_s}, \underbrace{0, \dots 0}_{D-M_s}), \tag{SI.3.1}$$

where $M_r, M_s \leq D$. Finally, we allow the kernel to have less features then the ambient space so that it does not express the whole $\mathbb{R}^D$: $K(\mathbf{x}, \mathbf{x}') = \frac{1}{M} \sum_{\rho=1}^{M} \mathbf{x}_\rho \mathbf{x}'_\rho$ where $M \leq D$. Therefore, we have 6 parameters: 1) $M_r, M_s$ representing how many directions training and test distributions have non-zero variance on, 2) $N$ quantifying how many directions target depends on and $M$ how many directions kernel can represent, 3) $\sigma^2, \tilde{\sigma}^2$ respectively the variances of training and test distributions.

Then plugging the parameters of this setting in Eq.(4), the generalization error simplifies to:

$$\kappa' = \frac{1}{2}\left[\left(1 + \tilde{\lambda} - \alpha\right) + \sqrt{\left(1 + \tilde{\lambda} + \alpha\right)^2 - 4\alpha}\right], \quad \tilde{\lambda} = \frac{\lambda}{\sigma^2 \min(1, M_r/M)}, \tag{SI.3.2}$$

$$\gamma = \frac{\alpha}{(\kappa' + \alpha)^2},$$

$$E_g = \frac{N_{rs}}{N_r} \frac{\gamma}{1-\gamma}\left(\frac{\tilde{\sigma}^2}{\sigma^2}\varepsilon^2 + \frac{\tilde{\sigma}^2 \kappa'^2}{(\alpha + \kappa'^2)}\sum_{\rho=1}^{N_r}\beta_\rho^2 + \tilde{\sigma}^2\sum_{\rho=N_r+1}^{M_r}\beta_\rho^2\right) + \frac{\tilde{\sigma}^2 \kappa'^2}{(\alpha + \kappa'^2)}\sum_{\rho=1}^{N_{rs}}\beta_\rho^2 + \tilde{\sigma}^2\sum_{\rho=N_{rs}+1}^{M_s}\beta_\rho^2, \tag{SI.3.3}$$

where we defined $\alpha = P/N_r$, $N_r = \min\{M, M_r\}$ and $N_{rs} = \min\{M, M_r, M_s\}$. Hence, the learning rate is determined by the minimum between number of features and the number of nonzero variance directions in the training distribution. Although complicated looking, Eq.(SI.3.2) predicts several interesting phenomena:

1. When there is an out-of-RKHS component in the target function, those components act like noise causing the effective noise given by:

$$\tilde{\varepsilon}^2 = \tilde{\sigma}^2 \frac{N_{rs}}{N_r}\left(\frac{\varepsilon^2}{\sigma^2} + \sum_{\rho=N_r+1}^{M_r}\beta_\rho^2\right). \tag{SI.3.4}$$

Hence, even there is no label noise in the training set, we observe a double-descent due to the effective noise.

Our theory suggests that the only way avoiding the contribution to noise from out-of-RKHS components is to not train on those directions, i.e. setting $M_r \leq M$. This has been demonstrated in Figure SI.3.1(a).

2. When the target depends on all dimensions, the irreducible error (last term) can be avoided if $M_s < N_r$. This simply means that only testing on the features which are not learned (either due to not training on those directions or the kernel not expressing them as features) causes an irreducible error in $E_g$. This has been demonstrated in Figure SI.3.1(b). Note that there is still double-descent since we are still training on the directions kernel does not express.

3. If the irreducible error is avoided by not testing on the directions where the training distribution has zero variance or kernel does not express ($M_s \leq M, M_r$), the generalization error approaches to 0 as $P \to \infty$ even though the target has out-of-RKHS components (See Figure SI.3.1(b)).

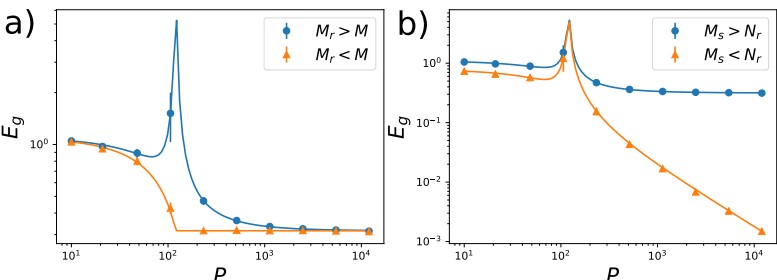

Figure SI.3.1: Effect of training and test distributions for out-of-RKHS target functions. The error bars indicate standard deviation over 30 averages.

## SI.4 Rotation Invariant Kernels and Neural Tangent Kernel

Another application of our theory is the study of rotation invariant kernels on high-dimensional input spaces. This type of kernels includes many popular kernels including Laplace kernels, radial basis function kernels and neural tangent kernel (NTK). Specifically, the kernel only depends on the inner product of two inputs: $K(\mathbf{x}, \mathbf{x}') = K(\mathbf{x} \cdot \mathbf{x}')$. In this case Mercer's decomposition takes the form:

$$K(\mathbf{x}, \mathbf{x}') = \sum_{k,m,n} \eta_{k,m,n} R_n(\|\mathbf{x}\|) R_n(\|\mathbf{x}'\|) Y_{k,m}(\mathbf{x}) Y_{k,m}(\mathbf{x}') \tag{SI.4.1}$$

where $Y_{km}$ are the hyper-spherical harmonics [55] which only depend on the angular coordinates of $\mathbf{x}$ and $R_n$ depend on the norm of the input which are usually orthonormal polynomials of order $n$. If the kernel is rotation invariant, the eigenvalues only depend on the degree of the spherical harmonics $k$ but not on the azimuthal coordinates $m$ which means $N(D, k) \sim \mathcal{O}(D^k)$ times degeneracy for each $\eta_{k,m,n} = \eta_{k,n}$. Note that $\eta_{k,n} \sim \mathcal{O}(N(D, k)^{-1})$ for kernel to have finite trace. Then we define $\mathcal{O}_D(1)$ quantity $\bar{\eta}_{k,n} \equiv N(D, k)\eta_{k,n}$.

Furthermore, we assume that the target function $\bar{f}(\mathbf{x}) = \sum_{k,m,n} \bar{a}_{k,m,n} R_n(\|\mathbf{x}\|) Y_{k,m}(\mathbf{x})$ has finite $L^2$ norm which implies that $\bar{a}_{k,n}^2 \equiv \frac{1}{N(D,k)} \sum_m \bar{a}_{k,m,n}^2$ is finite for each mode $k$. Note that the overlap matrix

$$\mathcal{O}_{kmn;k'm'n'} = \int d\mathbf{x}\, \tilde{p}(\mathbf{x}) R_n(\|\mathbf{x}\|) R_{n'}(\|\mathbf{x}\|) Y_{k,m}(\mathbf{x}) Y_{k',m'}(\mathbf{x})$$

is in general very difficult to calculate analytically without assuming specific probability distributions. To simplify, we consider probability distributions on hyperspheres with radius $R$ and $\tilde{R}$ for training and test distributions, respectively. Then considering the limit $P, D \to \infty$ while keeping $\alpha_k \equiv P/N(D, k)$ finite, we find that the different degree $k$ modes decouple over angular indices leading:

$$E_g = \frac{\gamma'}{1-\gamma} \left( \varepsilon^2 + \sum_{k'>k} \sum_n \bar{a}_{k',n}^2 \right) + \kappa^2 \sum_{n,n'} \frac{\bar{a}_{k,n}}{\alpha_k \bar{\eta}_{k,n} + \kappa} \left( \mathcal{O}_{nn'} + \frac{\gamma'}{1-\gamma}\mathbf{I} \right) \frac{\bar{a}_{k,n'}}{\alpha_k \bar{\eta}_{k,n'} + \kappa}$$
$$+ \sum_{k'>k} \sum_{nn'} \mathcal{O}_{nn'} \bar{a}_{k',n} \bar{a}_{k',n'}, \tag{SI.4.2}$$

where $\mathscr{O}_{nn'} = \int d\mathbf{x}\, \tilde{p}(\mathbf{x}) R_n(\|\mathbf{x}\|) R_{n'}(\|\mathbf{x}\|)$ and

$$\kappa = \lambda + \sum_{k'>k,n} \bar{\eta}_{k'n} + \kappa \sum_n \frac{\bar{\eta}_{k,n}}{\alpha_k \bar{\eta}_{k,n} + \kappa},\ \gamma = \alpha_k \sum_n \frac{\bar{\eta}_{k,n}^2}{(\alpha_k \bar{\eta}_{k,n} + \kappa)^2},\ \gamma' = \alpha_k \sum_n \frac{\mathscr{O}_{nn} \bar{\eta}_{k,n}^2}{(\alpha_k \bar{\eta}_{k,n} + \kappa)^2}.$$

$$\text{(SI.4.3)}$$

We notice that the effective regularization becomes $\tilde{\lambda} \propto \lambda + \sum_{k'>k,n} \bar{\eta}_{k'n}$ implying that the inductive bias of the kernel machine solely depends on the training distribution and can be altered by changing it. Furthermore, the target power for $k' > k$ acts as an effective noise and this is in fact an example of out-of-RKHS generalization: in the limit $P, D \to \infty$ for finite $\alpha_k \equiv P/N(D, k)$, the modes larger than $k$ are not in the sub-RKHS defined by polynomials of degree $k$. This has also been pointed out in [16]. There is also an irreducible error due to the target power for $k' > k$ which depends on both training and test distribution.

To conclude this section, we finally consider ReLU NTK regression with arbitrary depth. Note that the ReLU networks are homogeneous maps with respect to the norm of inputs [56]:

$$K(\mathbf{x}, \mathbf{x}') = \|\mathbf{x}\|\|\mathbf{x}'\| k\left(\frac{\mathbf{x} \cdot \mathbf{x}'}{\|\mathbf{x}\|\|\mathbf{x}'\|}\right),$$

$$\text{(SI.4.4)}$$

and therefore we can drop $n$-indices in the computation above. Notice that the self-consistent equation $\kappa$ in this case can be solved exactly and the solution is very similar to the one for linear regression: $\kappa' = \kappa/\bar{\eta}_k = \frac{1}{2}\left[(1 + \tilde{\lambda}_k - \alpha_k) + \sqrt{(1 + \tilde{\lambda}_k + \alpha_k)^2 - 4\alpha_k}\right]$ where the effective regularization is $\tilde{\lambda}_k = (\lambda + \sum_{k'>k} \bar{\eta}_{k'})/\bar{\eta}_k$. In this case the learning rate is controlled by the degeneracy of mode $k$: $\alpha = P/N(D, k)$. Homogeneity of the NTK Eq.(SI.4.4) implies that for inputs restricted to a $D$-sphere of radius $R$, the eigenvalues simply are multiplied by the norm squared: $\eta_k \to R^2 \eta_k$. Therefore when an NTK regression is performed on a training set with radius $R$ and tested on a sphere with radius $\tilde{R}$, the overlap matrix simply becomes diagonal with components $\frac{\tilde{R}^2}{R^2}$ and the analysis for linear regression can be directly applied here:

$$E_g^k = \tilde{R}^2 \left( \frac{\tilde{\varepsilon}^2}{R^2} \frac{\alpha_k}{(\kappa' + \alpha_k)^2 - \alpha} + \frac{\kappa'^2}{(\kappa' + \alpha_k)^2 - \alpha_k} \right) + \frac{\tilde{R}^2}{R^2} \sum_{k'>k} \bar{a}_k^2,$$

$$\text{(SI.4.5)}$$

where we define effective noise to be $\hat{\varepsilon}^2 = \varepsilon^2 + \sum_{k'>k} \bar{a}_k^2$. $E_g^k$ is the generalization error in learning stage $k$ for large $P, D$ limit and shows that it is very similar to linear regression we studied in Section 4. At each learning stage $k$, the kernel machine learns the $k^{th}$ mode and the higher modes act as irreducible error given by $\frac{\tilde{R}^2}{R^2} \sum_{k'>k} \bar{a}_k^2$. This implies that the irreducible error is controlled by the ratio of radii of the test and target distributions. Note that for larger test radius, generalization error increases for large width neural networks. We demonstrate this in Figure SI.4.1 where the inputs are randomly drawn from a sphere of radius $R$ for training set and test inputs are drawn from a sphere of radius $\tilde{R}$ for 2-layer NTK regression.

## SI.5 Interpolation vs. Extrapolation

Another subject where understanding OOD distribution is crucial is extrapolation from training data to test data whose support lies outside the training distribution. Our theory can also explain how kernel regression generalizes in extrapolation tasks for simple models like linear regression and band-limited Fourier kernels.

To understand extrapolation in linear regression, we introduce rectangular distributions for each direction $x_\rho$ defined as:

$$p(\mathbf{x}) = R_{\sigma_1}(x_1) \dots R_{\sigma_D}(x_D), \quad R_{\sigma_\alpha}(x) = \begin{cases} \frac{1}{2\sqrt{3}\sigma_\alpha} & -\sqrt{3}\sigma_\alpha \leq x \leq \sqrt{3}\sigma_\alpha \\ 0 & \text{otherwise} \end{cases}$$

$$\tilde{p}(\mathbf{x}) = R_{\tilde{\sigma}_1}(x_1) \dots R_{\tilde{\sigma}_D}(x_D).$$

$$\text{(SI.5.1)}$$

Then one can take $\tilde{\sigma}_\alpha^2 > \sigma_\alpha^2$ to change the support of train and test distributions. Note that the Gaussian measure is an example of interpolation since the support of the data is always $\text{Supp}(p(\mathbf{x})) =$

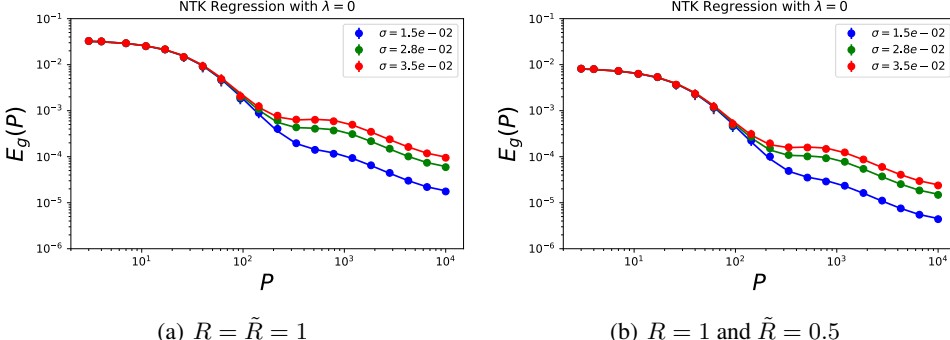

(a) $R = \tilde{R} = 1$           (b) $R = 1$ and $\tilde{R} = 0.5$

Figure SI.4.1: 2-Layer NTK Regression on random spherical data. a) The training and test distribution radii are the same $R = \tilde{R} = 1$. b) Training distribution radius is $R = 1$ and test distribution radius is $\tilde{R} = 0.5$. As predicted by theory, $E_g$ is lower in the former case.

$(-\infty, \infty)$. One can easily show that the solution to the integral eigenvalue problem for kernel is the same as when the distributions are Gaussian with diagonal covariance matrices: the kernel eigenvalues are given by $\eta_\rho = \sigma_\rho^2/D$ and eigenfunctions are $\phi_\rho(\mathbf{x}) = x_\rho/\sigma_\rho$ leaving the features unchanged. Then the analysis in Section 4 exactly applies to the extrapolation in this scenario implying that the extrapolation and interpolation are the same when linear regression for linear tasks are concerned. For NTK, this finding has been stated in [57] that as long as all $x_\rho$ with nonzero power in target is expressed in the kernel, the kernel regression should be able to extrapolate. This intuitively makes sense since once the parameters are $\beta$ are found, the extrapolation should be trivial.

However, with nonlinear features we find that learning nonlinear functions, although possible, is much more costly when extrapolating with rectangular distributions than Gaussian distributions. Heuristically, we claim that the Gaussian distribution has always the same support and can still be thought of as interpolation no matter how much the variance is changed, while in the rectangular case supports for training and test distributions may be different and the kernel machine might not be trained on the region it is tested on. Here we demonstrate an example to explain why this is the case.

We consider a band-limited kernel with Fourier features in 1D, $K(x, x') = \sum_{k=1}^{N} \cos k\pi(x - x')$ and a periodic target function $\bar{f}(x) = e^{\cos(\pi x - \theta)} + e^{\cos(\pi x + \theta)}$ centered around its mean. For the inputs, we consider centered Gaussian distributions with varying variances and rectangular distributions on the interval $x \in [-a, a]$ with varying $a$. By $\alpha = P/N$, we denote the ratio of training samples to the number of features and for certain values of $\alpha$ we compare the estimator (Eq.(SI.1.52)) to the target function $\bar{f}$. In Figure SI.5.1(a,b), we find that the estimator perfectly matches the target when $\alpha = 4$ for both narrow and wide Gaussian training distributions. On the contrary, when the rectangular distributions are used for training in Figure SI.5.1(c,d), we find that interpolation is achieved as soon as $\alpha = 1$ while extrapolation requires much more samples $\alpha = 250$. We attribute this behavior to the observation that some eigenvalues in the rectangular distribution case effectively goes to 0 as can be seen from Figure SI.5.1(f) while for Gaussian distribution they stay large Figure SI.5.1(e). This means that as the range of the rectangular distribution gets smaller, more modes in the target function become out-of-RKHS leading to an irreducible error.

## SI.6   Numerical Methods

We performed our experiments using JAX software [30] and NeuralTangents package [29] on Google Colaboratory environment [58]. Specifically, the automatic differentiation capabilities of JAX helped us optimize over training and test distributions on MNIST digits [59]. All code used to perform experiments and generate figures can be accessed at https://github.com/Pehlevan-Group/kernel-ood-generalization.

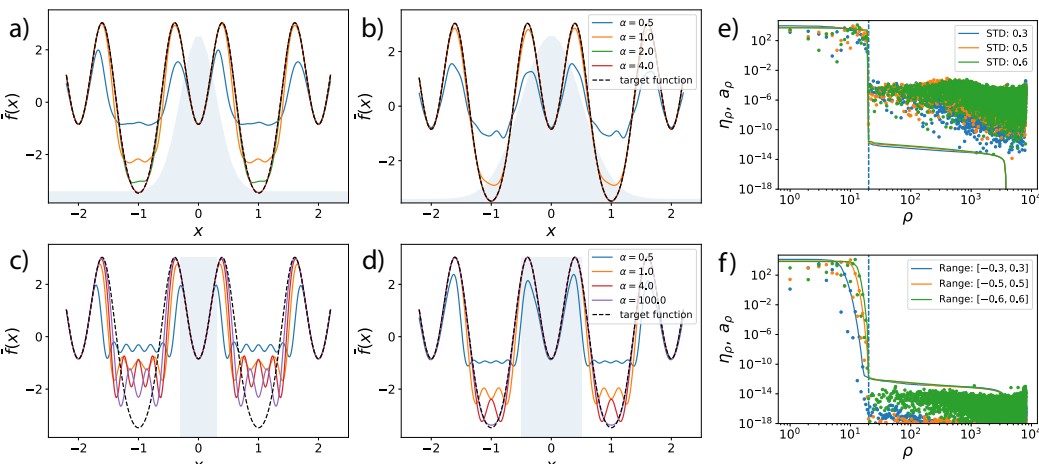

Figure SI.5.1: Interpolation with Gaussian distribution (first row) vs. Extrapolation with Rectangular distribution (second row). The estimator obtained from kernel regression is presented. a, b) Interpolation with narrower width (a) requires more training samples than wider widths (b). c, d) In comparison, rectangular distributions work are able to interpolate well (at $\alpha = 1$) while extrapolation takes much more samples ($\alpha \sim 100$) to extrapolate. (e,f) The kernel eigenvalues on training distribution and target power are shown for Gaussian and rectangular distributions, respectively. Dashed lines indicate the number of features $N$ represented in the kernel. We observe that for varying Gaussian distribution widths, the spectrum does not change significantly while for rectangular case some eigenvalues effectively go to 0 implying an irreducible error in the generalization.