# OpenReview forum: "Out-of-Distribution Generalization in Kernel Regression"
_NeurIPS.cc/2021/Conference — NeurIPS 2021 Poster_

### Official Review · Reviewer_wKiN · 2021-07-05

**Rating:** 6
**Confidence:** 4

**Summary:**

This paper proposes asymptotic convergence results of kernel regression in the out-of-distribution setting via replica tricks. Some findings and examples are given.

**Limitations And Societal Impact:**

it's enough.

**Main Review:**

I’m not an expert on statistical mechanics/physics though I know replica methods used in the machine learning community. I haven't checked the proofs in the appendix and will just trust authors in the correctness of the proof.

**Main issues:**

Nevertheless, this paper corresponds to eigenvalue decomposition for $R^M \times R^M$ matrix (with infinite $M$ and infinite feature dimension $D$) is still questionable to me.
As indicated by random matrix theory, matrix in spectral norm in high dimensions is quite different from the low-dimensional case. Besides, this paper does not strictly distinguish the difference between the integral operator and kernel matrix. In fact, such difference would be non-negligible: “The large eigenvalues of kernel matrix approximate the corresponding eigenvalues of kernel operator very well. The small eigenvalues are away from the smallest eigenvalues of kernel operator, but still relatively large.” Refer to the following paper:

*Ma, C., Wu, L. and Weinan, E., 2020. The slow deterioration of the generalization error of the random feature model. Mathematical and Scientific Machine Learning, 2020.*

Proposition 1 is not enough and I suggest the authors to provide further analysis. For example, quantitative analysis/discuss the gap between the training/test distribution via some metric, e.g., MMD or total variance norm, and then study how such gap effects the third term in Eq. (7). It would be nice to provide error bounds in probability.

In Section 4, the authors consider linear regression with the linear kernel for Gaussian data, and thus the kernel eigenvalue equation can be solved exactly. This is a common way in replica methods. But this appears a little “overclaim” on application to any data, any kernel, any distribution throughout the paper.


**Minor issues:**

All kernel eigenvalues are assumed to be non-zero, which means the RKHS is dense in L^2(X). This paper assumes the target function \bar{f} lies in a RKHS, which is fair and standard in learning theory. In this case, if my understanding was right, “N” should be “M” in line 86?

It would be better if the definitions of some notations and expressions follow the machine learning community. For example, “generalization error” -> “excess risk” in line 58. They are in fact different and have different meanings in learning theory. Besides, I suggest the authors to add “P” in the loss term in Eq.(1) as the empirical error:
$$\frac{1}{2P} \sum_{\mu=1}^P (f(x^\mu) - y)^2 + \lambda || f ||_{\mathcal{H}}^2$$

This setting is common in statistical learning theory though [14,15] is without $P$.
With $P$, it is natural to assume that $\lambda = O(P^{-a})$ with $a \in (0,1]$, refer to

*Cucker, Felipe, and Ding Xuan Zhou. Learning theory: an approximation theory viewpoint. Vol. 24. Cambridge University Press, 2007.*


Typo:

The test propability meansure should be $\tilde{p}(x)$ in line 135.

The transpose operation is missing for $\Phi$ in Eq. (SI.4).


**Time Spent Reviewing:**

3

---

> ### Author Response · Authors · 2021-08-10
> **Response to Reviewer 'wKiN'**
>
> ***Main issues***
>
> *Nevertheless, this paper corresponds to eigenvalue decomposition for $\mathbb{R}^M\times\mathbb{R}^M$ matrix (with infinite $M$ and infinite feature dimension $D$) is still questionable to me. As indicated by random matrix theory, matrix in spectral norm in high dimensions is quite different from the low-dimensional case. Besides, this paper does not strictly distinguish the difference between the integral operator and kernel matrix. In fact, such difference would be non-negligible: “The large eigenvalues of kernel matrix approximate the corresponding eigenvalues of kernel operator very well. The small eigenvalues are away from the smallest eigenvalues of kernel operator, but still relatively large.” Refer to the following paper:*
>
> *Ma, C., Wu, L. and Weinan, E., 2020. The slow deterioration of the generalization error of the random feature model. Mathematical and Scientific Machine Learning, 2020.*
>
> We thank the reviewer for this great question. We actually do make a distinction between the integral operator and the (empirical) kernel matrix. To calculate generalization error using Eq. (7), one should compute the integral operator eigenvalues and eigenfunctions for a given  kernel and data distribution. We will clarify this point with a citation to the paper linked.
>
> We realized our discussion of how we apply our equations to real data around line 135 may cause a confusion. Here, we state that we assume an empirical (atomic) distribution, $p(\mathbf{x})=\sum_{\mu=1}^P p_{\mu} \delta(\mathbf{x}-\mathbf{x}^\mu)$ and the kernel eigenvalue problem for the integral operator reduces to computing the eigenvalues and eigenvectors of the $P\times P$ kernel matrix. Here $P$ actually refers to the size of the whole training set and not the $P$ training samples as a variable as in the rest of the paper. In other words, in this case, we are actually calculating the integral operator's eigenvalues. We will change the notation here to make this distinction.
>
> A subtle distinction between the eigenvalues of the integral kernel operator and the empirical kernel matrix appears in our calculation of the generalization error, and is in a sense the reason for the application of the replica method. When calculating the predictor for a given dataset, $f^*$, one needs to invert the empirical kernel matrix. This matrix is dataset dependent. To calculate the average generalization error over all possible realizations of the training datasets (and hence empirical kernel matrices), we resort to the replica method, treating the each realization as a "quenched disorder". The eigenvalues of the empirical kernel does not appear in Eq. (7) because of this averaging.
>
> *Proposition 1 is not enough and I suggest the authors to provide further analysis. For example, quantitative analysis/discuss the gap between the training/test distribution via some metric, e.g., MMD or total variance norm, and then study how such gap effects the third term in Eq. (7). It would be nice to provide error bounds in probability.*
>
> We thank the reviewer for his/her assessment and the great suggestion. We will look for an interpretation of the third term in Eq. (7) through some metric of distributional difference. If we find any, we will include that in the final version.
>
> However, we do like to note that the particular form presented in Eq.(7) is intended to show how in-distribution generalization analytically differs from out-of-distribution generalization by the third term, and $\mathcal{O}'$ is our intended "matrix valued" metric which becomes $0$ when the distributions match.
>
> *In Section 4, the authors consider linear regression with the linear kernel for Gaussian data, and thus the kernel eigenvalue equation can be solved exactly. This is a common way in replica methods. But this appears a little “overclaim” on application to any data, any kernel, any distribution throughout the paper.*
>
> Thank you for your comment and giving us an opportunity to clarify the generality of our results. Our formula (Eq. 7) is indeed applicable to any kernel and any target function with any distribution (including empirical) given that one has to solve for the self-consistent equation for $\kappa$ and compute the generalization error (Eq. 7) numerically. We had demonstrated this in several examples throughout the paper including regression with NTK (Figure 1) and neural network experiments (Figure 2), and real datasets (MNIST). Furthermore, previous works which use similar methods also showed good agreement with experiments, specifically in [14,15]. It is in this sense that we claim our model is generally applicable. We study linear regression because it is special in that $\kappa$ is exactly solvable and provides further analytical tractability.
>
> Since we do not want to mislead our readers, we will emphasize the numerical solution caveat to the generality of our results when presenting Eq. (7) and in our discussion.
>
>
> ***Minor issues***
>
> *All kernel eigenvalues are assumed to be non-zero, which means the RKHS is dense in $L^2(X)$. This paper assumes the target function $\bar{f}$ lies in a RKHS, which is fair and standard in learning theory. In this case, if my understanding was right, “N” should be “M” in line 86?*
>
> We thank the reviewer for carefully reading our manuscript. In our notation, $N$ denotes the number of features in the kernel (number of non-zero eigenvalues) and $M$ denotes the number of eigenfunctions which span $L^2(\mathbb{R}^D)$; we consider generic targets which have weights on all $M$ eigenfunctions. In main text, for presentational simplicity, we assume non-zero eigenvalues and these two coincide ($N = M$).
>
> However, in SI, we also allow target functions which do not lie in the RKHS. Starting from line 738, we consider the case where some eigenvalues are zero (e.g. $N < M$) and in that case we show both analytically and experimentally that our theory holds in this special case where target function has power on the eigenmodes which are not expressed in the kernel.
>
> Indeed in line 86, the $N$ should be $M$, and this is a typo. However, as we explained above, our theory is more general. In the case where target function has power on the eigenmodes which are not expressed in the kernel, these modes contribute to the generalization error as irreducible error and noise on labels as described below equation SI.65 on line 741.
>
> We will make these points more clear.
>
> *It would be better if the definitions of some notations and expressions follow the machine learning community. For example, “generalization error” -> “excess risk” in line 58. They are in fact different and have different meanings in learning theory.*
>
> Thank you for the suggestion. Just to clarify, we use the convention in [2] (Mohri, Rostamizadeh, Talwalkar, Foundations of Machine Learning) for naming generalization error.
>
> *Besides, I suggest the authors to add “P” in the loss term in Eq.(1) as the empirical error:*
>
> \begin{align}
>     \frac{1}{2P}\sum_{\mu = 1}^P (f(x^\mu) -y^\mu)^2 + \lambda||f||_\mathcal{H}^2
> \end{align}
>
> *This setting is common in statistical learning theory though [14,15] is without $P$. With $P$, it is natural to assume that $\lambda \sim \mathcal{O}(P^{-a})$ with $a \in (0,1]$, refer to*
>
> *Cucker, Felipe, and Ding Xuan Zhou. Learning theory: an approximation theory viewpoint. Vol. 24. Cambridge University Press, 2007.*
>
> Thank you for this suggestion. We will consider it.
>
> ***Typo:***
>
> *The test propability meansure should be
>  in line 135.*
>
> *The transpose operation is missing for  in Eq. (SI.4).*
>
> We thank the reviewer for catching these typos. We have fixed them.

---

### Official Review · Reviewer_Asg9 · 2021-07-07

**Rating:** 6
**Confidence:** 3

**Summary:**

The paper applies the so-called "replica method" to tackle the generalization issue for kernel regression where there is a distribution shift between training data and testing data. It presents the derivation of kernel regression by using an alternative perspective on RKHS functions. Empirical studies on real datasets including MNIST have been shown. Discussions with the neural tangent kernel are presented.

Despite the interesting idea, the clarity of the paper can be improved to better appreciate the ideas and presentations. Moreover, the significance and impact of the work is also unclear, making it difficult to appreciate the advancement from existing results.

**Limitations And Societal Impact:**

The limitation is mainly the clarity of presentation and bringing up the significance/advancement from existing work (detailed in the main review).

Social impact is not an issue here but more discussions on how the distribution shift can be connected to social issues, such as fairness machine learning or privacy-preserving concerns in machine learning can be useful.

**Main Review:**

The paper studies the generalization issue for kernel regression. It presents an alternative perspective on deriving the objectives as well as interpret the kernel.

Despite the interesting idea, the criticism on the paper is associated with the clarity of presentation and significance of impact to the community.

It is not easy to read and follow the ideas presented in the paper as it is not self-contained. Even after reading the appendix, it is still unclear what is "replica method" and how is this useful to improve the overall learning scheme for kernel regression.
more ideas are introduced in the Experimental section which induces more doubts and concerns. For instance, the beneficial and adversarial matrix in Fig 1 b): is this related to adversarial attack during learning procedure?
Fig 3 b) shows the double descent phenomenon while not clearly explained.

In terms of significance and impact, it is unclear how the overall setting improves the kernel regression when testing on cases with distribution shift. The regularised kernel-based regression objective is based on scalar-valued RKHS functions and squared loss. How does this approach objective different from learning a conditional mean embedding? With conditional mean embedding, distribution shift can be easily incorporated. Results on this approach have been studied in
 Grunewalder et, al. "Conditional mean embeddings as regressors"
Moreover, Fukumizu et,al. "Kernel Bayes rule" is another useful reference in terms of manipulating conditional distributions and joint distribution in the kernel setting.
I am curious how the proposed approach compares with the existing ones.
In addition, is the result only valid for energy-based models like Gibbs distribution presented? it will be nice to discuss what may happen beyond.

minor issues:

- \langle \rangle are used for both RKHS norm (e.g. Eq(1)) as well as the norm w.r.t. distribution/samples (e.g. Eq(2)) which are incoherent.

- sample size notation P is non-conventional and especially can be confusing when distribution p(x) is used.

- in Eq.(6), is Z the normalizer/partition function? otherwise, function O can be better explained and notated.

**Time Spent Reviewing:**

4

---

> ### Author Response · Authors · 2021-08-10
> **Response to Reviewer 'Asg9' Part 1/2**
>
> ***Main Review***
>
> *The paper studies the generalization issue for kernel regression. It presents an alternative perspective on deriving the objectives as well as interpret the kernel.*
>
> *Despite the interesting idea, the criticism on the paper is associated with the clarity of presentation and significance of impact to the community.*
>
> We thank you for your criticism, which made use realize that some of the choices we made in writing our paper obscured our points for readers with different backgrounds. We apologize. Below we address the specific points you have raised. To summarize, we will implement the following changes to clarify our presentation and emphasize the significance of our results:
> - Make a clearer and more intuitive connection between kernel regression and statistical physics techniques we use.
> - Provide more details about the replica method and its use.
> - Clarify the derivation of the formula for generalization error and how it differs from previous works, especially [14,15], where the training and test distributions match. When training distributions don't match, our new theory gives a generalization error which is the sum of the error from [14,15] plus a quadratic form on the overlap matrix.
> - Explain the need for a precise theory of generalization in kernel regression rather than error bounds when there is distribution mismatch.
> - Discuss our experimental methods and findings more clearly.
>
> We kindly ask you to reconsider your score based on our proposed changes and clarifications, and our responses below.
>
> *It is not easy to read and follow the ideas presented in the paper as it is not self-contained. Even after reading the appendix, it is still unclear what is "replica method" and how is this useful to improve the overall learning scheme for kernel regression.*
>
> We thank the reviewer for his/her constructive comments. We apologize for the confusion our presentation caused. While replica method is a widely used technique in statistical mechanics and has found many applications in machine learning theory [14,15,19,21,30,35-38], indeed our current introduction of the replica method is very brief. Around Equation SI.14 and below we describe the mechanics of the replica method but not the philosophy and reasoning behind it. We will provide an extended discussion in the SI and a summary in the main text below equation (6).
>
> We propose the following text around equation 6 (also see related Remark 2 in line 115 of the current manuscript; we will also add more of the classical references for the use of replica method in machine learning):
>
> "We map this problem to statistical mechanics by defining a Gibbs distribution $\propto e^{-\beta H(\mathbf{w};\mathcal{D})}$ over estimator weights $\mathbf{w}$ which concentrates around the kernel regression solution $\mathbf{w}^*$ as $\beta\to\infty$. This can be used to calculate any function  $O(\mathbf{w}^*;\mathcal{D})$ by the following trick:
>
> $ O(\mathbf{w}^*;\mathcal{D}) = \lim_{\beta\to\infty}\frac{\partial}{\partial J} \log Z[J;\beta, \mathcal{D}]\big|_{J=0} ,\quad Z[J;\beta, \mathcal{D}] = \int d\mathbf{w} e^{-\beta H(\mathbf{w};\mathcal{D}) + J O(\mathbf{w})}. \qquad (6)$
>
> Therefore, the dataset average $\langle O(\mathbf{w}^*;\mathcal{D})\rangle_\mathcal{D}$ requires averaging the logarithm of the partition function $\langle\log Z\rangle_\mathcal{D}$.  Further, experience from the study of physics of disordered systems suggests that the logarithm of the partition function concentrates around its mean (is self-averaging) for large $P$ [13], making our theory applicable to the typical case. However, this average is analytically hard to calculate due to the partition function appearing inside the logaritm. To proceed, we resort to the replica method from statistical physics [13], which uses the equality $\langle\log Z \rangle_\mathcal{D} = \lim_{n\rightarrow 0}\frac{\left<Z^n\right>_\mathcal{D}-1}{n}$. The method proceeds by calculating the right hand side for integer $n$, analytically continuing the resulting expression to real valued $n$, and performing the limit. While non-rigorous, the replica method has been succesfully used in the study of the physics of disordered systems [13] and machine learning theory [14,15,19,21,30,35-38]. A crucial step in our computation is approximating $\mathbf{w}^\top\mathbf{\Psi}(\mathbf{x})$ as a Gaussian random variable via its first and second moments when averaged over the training distribution $p(\mathbf{x})$. It has been shown that this approximation yields perfect agreement with experiments [27,15,16]]. Details of our calculation is given in SI Section.1. "
>
> After this clarification, we want to point that we are not using the replica method to improve the overall learning scheme of kernel regression. Our goal is to obtain a predictive analytical theory of kernel regression as is using the replica method. Our work can be employed to understand various phenomena in kernel regression learning curves when there is mismatch between training and test distributions.
>
> *more ideas are introduced in the Experimental section which induces more doubts and concerns. For instance, the beneficial and adversarial matrix in Fig 1 b): is this related to adversarial attack during learning procedure? Fig 3 b) shows the double descent phenomenon while not clearly explained.*
>
> After computing an analytical expression for kernel regression, we presented several examples in the Experimental sections to (1) show the theory matches the experiments, (2) to display how it can be applied engineering training sets and (3) to analyze how testing on a different distribution than training affects generalization error. We will edit these sections to improve clarity. We address the specific points raised by the reviewer below. We would appreciate other pointers.
>
> In Figure (1b), matrices shown are the overlap matrices between training and test distributions where the test distribution has more weights on beneficial/adversarial examples, respectively. In that experiment, training distribution is fixed and it is not related to adversarial attacks during the learning. Our formula allows us, through gradient descent(ascent), to extract which test distributions are beneficial(detrimental) for generalization. A similar analysis can be done when the test distribution is fixed and it is studied in Figure (2) for neural networks and NTK. However, we apologize for the bad choice of wording since the samples are not adversarial in the usual sense used in machine learning community and will replace it with "detrimental" in our next draft.
>
> Regarding the double-descent features in Fig. 3, we agree that it needed more explanation and we are grateful that the reviewer pointed it out. The double-descent observed in our manuscript is of the same nature as "Nakkiran - More Data Can Hurt for Linear Regression (2019)" and called sample-wise double-descent. We found in our analytical expressions and experiments that the location and the size of the double-descent peak can be manipulated by changing both training and test distributions. Specifically, in linear regression, the location of double-descent peak directly depends on the training distribution as stated in line 232. Furthermore, the size of double-descent can be altered by effective regularization given in line 210 and can be mitigated by changing the training distribution. We will improve our explanation of these facts and discuss the implications of our results in more depth during the revision.

---

> ### Author Response · Authors · 2021-08-10
> **Response to Reviewer 'Asg9' Part 2/2**
>
> *In terms of significance and impact, it is unclear how the overall setting improves the kernel regression when testing on cases with distribution shift. The regularised kernel-based regression objective is based on scalar-valued RKHS functions and squared loss. How does this approach objective different from learning a conditional mean embedding? With conditional mean embedding, distribution shift can be easily incorporated. Results on this approach have been studied in Grunewalder et, al. "Conditional mean embeddings as regressors" Moreover, Fukumizu et,al. "Kernel Bayes rule" is another useful reference in terms of manipulating conditional distributions and joint distribution in the kernel setting. I am curious how the proposed approach compares with the existing ones. In addition, is the result only valid for energy-based models like Gibbs distribution presented? it will be nice to discuss what may happen beyond.*
>
> We thank the reviewer for these comments and pointers. There are multiple points raised here, which we would like to address separately.
>
> - We first would like to note that our aim here is to study how generalization error is affected by a distribution shift rather than providing a tool for improving generalization error. We demonstrate how a distribution mismatch may improve generalization as an example of our results.
>
> - While it is true that our theory works for scalar-valued RKHS functions, it can be extended to vector functions/targets given that its components/classes are uncorrelated in the scalar-RKHS. Then each unit in the output can be treated separately as a scalar function and hence the optimization problem can be decomposed into minimizations of the corresponding losses for each class (see eq.(27) in [14]). For example, Neural Tangent Kernel corresponding to infinite width limit of feedforward neural networks is of this class [11].
>
> - We thank the reviewer for bringing up Conditional mean embedding. We believe the reviewer is asking two separate clarifying questions here, which we address.
>
> First is the difference between the kernel regression cost we use (Eq. 1) vs the vector-valued kernel regression problem that conditional mean embedding corresponds to  (Eq. 9 and 10 of Grunewalder et al.). While the solution for the former is a distribution function over outputs $\mu(\mathbf{x})(.): \mathcal{Y} \to \mathbb{R}$ for each input $\mathbf{x}$, in kernel regression the solution is simply a function in the RKHS $\mathcal{H} \ni f(\mathbf{x}): \mathcal{X} \to \mathbb{R}$ which maps inputs to outputs. Note that the conditional mean embedding necessitates the definition of another RKHS on the output space $\mathcal{Y}$ whose kernel is denoted as $L: \mathcal{Y}\times\mathcal{Y} \to \mathbb{R}$ in the paper by Grunewalder et. al (Eq.1) where the vector-valued kernel regression solution is stated as:
>
> $ \hat \mu(\mathbf{x})(.) = \sum_{\mu, \nu=1}^P L(.,\mathbf{y}^\mu) (\mathbf{K} + \lambda\mathbf{I})_{\mu\nu}^{-1}K(\mathbf{x}^\nu,\mathbf{x}), $
>
> where $\mathbf{K} = K(\mathbf{x}^\mu, \mathbf{x}^\nu) $ for $\mu,\nu = 1...P$ is the kernel Gram matrix and $\lambda$ is ridge parameter. Then it is clear that kernel regression simply corresponds to the special case where $L(\mathbf{y},\mathbf{y}') = \delta(\mathbf{y}-\mathbf{y}')$ such that the expectation value of the output $\mathbf{y}$ with respect to the the distribution $\langle\mathbf{y}, \hat\mu(\mathbf{x})(\mathbf{y}) \rangle_{L}$ reduces to the usual kernel regression solution we study here. Despite their relations to some degree, our work and the reviewer's references study different learning algorithms and attempt to solve different problems. Furthermore, an out-of-distribution analysis needs to be carried separately in this setting since the proposed regression problem by the reviewer only finds a conditional distribution on outputs given the training data and we believe evaluating the performance of this model on a different distribution still requires techniques like the one we employed here.
>
> Second is the question of what our paper contributes to beyond Grunewalder et al. This paper does not consider the case where test distribution differs from training distribution, while in this manuscript we study generalization under distribution shift. Further, they derive upper bounds on the generalization, while we obtain an analytical formula for the typical case (using statistical mechanics) which shows near-perfect agreement with experiments.
>
> In any case, we believe that attacking the out-of-distribution problem via functional methods such as kernel mean embedding is an interesting and potentially a promising idea, and we thank the reviewer for making this connection. We will review this connection in our discussion section with citations to the references the reviewer pointed to. We would be grateful if the reviewer points out if we are missing something here.
>
> - We thank the reviewer to bring up Kernel Bayes' Rule paper by Fukumize et. al. which studies Bayes' rule on kernel embeddings of probability distributions and inference using them. As we mentioned above, this is yet another technique to generalize to unseen data however it does not study out-of-distribution generalization and one still needs to employ techniques like ours to analyze such situations. Again, we would be grateful if the reviewer points out if we are missing something here.
>
> - The Gibbs measure is just a mathematical tool we introduce to map the problem to statistical mechanics. Our result is valid for general kernel regression.
>
> ***Minor issues***
>
> - *$\langle \rangle$ are used for both RKHS norm (e.g. Eq(1)) as well as the norm w.r.t. distribution/samples (e.g. Eq(2)) which are incoherent.*
>
> We thank the reviewer for bring this point up. While it seems incoherent, we distinguish them by the subscripts $\langle.,.\rangle_\mathcal{H}$ and $\langle.,.\rangle_{p(\mathbf{x})}$, respectively.
>
> - *sample size notation $P$ is non-conventional and especially can be confusing when distribution $p(x)$ is used.*
>
> We agree that denoting sample size to be $N$ is the more conventional way, however we use $N$, together with $M$, to be the number of features and the number of eigenmodes of the kernel, respectively. We apologize for inconvenience.
>
> - in Eq.(6), is $Z$ the normalizer/partition function? otherwise, function $O$ can be better explained and notated.
>
> In Eq.(6), $Z$ is indeed the partition function and $O$ is a functional of estimator of $f$ (observable) such as the generalization error or training error. Indeed we realize that $O$ is used for both observables and overlap matrix and we will fix it in the revised version of our manuscript. We thank the reviewer for pointing this out.

---

> > ### Comment · Reviewer_Asg9 · 2021-08-28
> > **Reviewer response**
> >
> > The reviewer appreciates the detailed explanation and clarifications from the author, which helps the understanding and appreciation of the manuscript to a greater extent.  The score is raised accordingly.

---

### Official Review · Reviewer_6Lb8 · 2021-07-17

**Rating:** 7
**Confidence:** 3

**Summary:**

This paper studies the out-of-distribution generalization error in kernel ridge regression (KRR) with any choice of regularization parameter $\lambda$ (including the case of interpolation), and as a general functional of the training and test distributions using the replica method in statistical physics. In addition,

-- they provide explicit closed-form expressions for the case of the linear model.

-- using the neural tangent kernel (NTK), they empirically evaluate their asymptotic formulas on modified samples of the MNIST test dataset to provide best-case training distributions, which can sometimes be different from the one that exactly matches.

**Limitations And Societal Impact:**

Yes, the authors have adequately addressed the limitations and potential negative societal impacts of their work.

**Main Review:**

The setting of transfer learning in kernel ridge regression is well-motivated and an important topic for research, especially in the context of the recent discovery of the NTK regime, which increases the relevance of kernel analyses in the context of modern ML. To the best of my knowledge, I have not seen precise asymptotic analyses of the transfer learning setting in KRR, and so I think this paper will be of interest to the theoretical ML community. The experiments done on MNIST with a 3-layer NTK are also interesting.

The paper would benefit from a more detailed comparison with the results already derived for the case where the training and test distributions are the same (e.g. citation [14] in the submission), and a discussion of the additional technical non-trivialities that arise in the analysis for the out-of-distribution case. Additionally, are there settings that are more general than the linear model where the expressions could potentially be made closed-form?

**Time Spent Reviewing:**

2

---

> ### Author Response · Authors · 2021-08-10
> **Response to Reviewer '6Lb8'**
>
>
> ***Main Review***
>
> *The setting of transfer learning in kernel ridge regression is well-motivated and an important topic for research, especially in the context of the recent discovery of the NTK regime, which increases the relevance of kernel analyses in the context of modern ML. To the best of my knowledge, I have not seen precise asymptotic analyses of the transfer learning setting in KRR, and so I think this paper will be of interest to the theoretical ML community. The experiments done on MNIST with a 3-layer NTK are also interesting.*
>
> We thank the reviewer for his/her comments and support.
>
> *The paper would benefit from a more detailed comparison with the results already derived for the case where the training and test distributions are the same (e.g. citation [14] in the submission), and a discussion of the additional technical non-trivialities that arise in the analysis for the out-of-distribution case. Additionally, are there settings that are more general than the linear model where the expressions could potentially be made closed-form?*
>
> We thank the reviewer for these suggestions. We agree a more detailed comparison with the in-distribution cases studied in [14,15] would improve the contrast with out-of-distribution scenarios in our work and we will improve the discussion section to fill that gap.
>
> At a high level, the first two terms of our generalization formula (Eq. 7) give the generalization error when test and train distributions are the same, and is identical to given in [14,15]. The third term captures the error arising from the difference, and is equal to zero when $\mathcal{O}=I$, i.e. when the test and train distributions are the same. Thus, in this limit, we recover the results of [14,15]. We will expand our discussion of this fact (currently in Remark 3, line 119).
>
> To see the main technical difference between the current work and [14,15], note that the estimator derived in [15] and here are the same since they only depend on the training set. The difference in calculation starts when computing the generalization error $\langle\big(f^*(\mathbf{x})-\bar f(\mathbf{x})\big)^2\rangle_{\tilde p(\mathbf{x})}$ with respect to a different distribution $\tilde p(\mathbf{x})$ starting at SI Section SI.1.5 at line 717. This results in the introduction of the overlap matrix defined in Eq. (8), which is ultimately related to the fact that the kernel eigenvalues and eigenfunctions differ with respect to different distributions.
>
>
> Finally, we have already provided one more example with a closed-form solution in supplementary material, SI.3. This is the case of rotation invariant kernels with data uniformly distributed on a D-sphere, and train and test distributions differ only by the radius of the sphere. In this case, due to rotational invariance, eigenfunctions are spherical harmonics. Eigenvalues are degenerate with all harmonics with the same degree (but different modes) having the same eigenvalue. The degeneracy scales with the input dimension as $\mathcal{O}(D^{k})$ where $k$ is the degree of the harmonic. In the limit $D, P \to \infty$, generalization error in each degenerate space reduces to the linear model. In SI Section.3, we detailed this solution and verified our predictions with experiments for NTK and Gaussian RBF kernels.
>
> Unfortunately we currently do not have any other models with closed-form solutions.

---

### Official Review · Reviewer_U6wb · 2021-07-20

**Rating:** 7
**Confidence:** 3

**Summary:**

This paper studies the generalization capabilities in kernel regression in the particular case where the distribution of training and testing is different.
They quantify the mismatch between distributions for a given kernel and show that its a key determinant of generalization performance.
They also propose how to optimize training and test distributions for a given data budget to find the best and worst case generalizations under a shift.
The authors investigate how in real data this hypothesis can be confirmed and show how learning rate with training set size changes with training distribution and cases where choosing it different than test distribution helps.
Their approach provides analytical tool for having more precise predictions.

**Limitations And Societal Impact:**

The authors adequately addressed the limitations and potential negative societal impact of their work.

**Main Review:**

It is a well-written and well-motivated paper. The problem of out-of-distribution generalization of neural models is a well-studied area and this work provides valuable insight in this area.

Questions for the authors:

- The effectiveness of this approach is in knowing what the test distribution look like. What do you think about the potential of the analysis you did for when the test distribution can be estimated but is not known? In NLP problems for instance, the distribution of output
 language can be to an extent estimated.

- You show on MNIST dataset that the beneficial measure has lower generalization error while the adversarial measure has higher generalization error than the matched case. You use gradient ascent to find adversarial test measures. Does the type of adversarial samples (as opposed to less or more challenging/natural cases) influence the conclusion?

- What are the limitations (computation-wise) of this approach? How well do you think it's adaptable for large (real world) datasets. In case of machine translation for example, the training data can be a couple of million samples.


**Time Spent Reviewing:**

4

---

> ### Author Response · Authors · 2021-08-10
> **Response to Reviewer 'U6wb'**
>
>
> ***Main Review***
>
> *It is a well-written and well-motivated paper. The problem of out-of-distribution generalization of neural models is a well-studied area and this work provides valuable insight in this area.*
>
> We thank the reviewer for his/her support, suggestions and time.
>
> *Questions for the authors:*
> - *The effectiveness of this approach is in knowing what the test distribution look like. What do you think about the potential of the analysis you did for when the test distribution can be estimated but is not known? In NLP problems for instance, the distribution of output language can be to an extent estimated.*
>
> We thank the reviewer for this very interesting question, which we will discuss in the final version. We believe, though did not test, that our theory could work by estimating test distribution $\tilde p(\mathbf{x})$ through density estimation methods such as kernel density interpolators. This would be an interesting future work.
>
>
> - *You show on MNIST dataset that the beneficial measure has lower generalization error while the adversarial measure has higher generalization error than the matched case. You use gradient ascent to find adversarial test measures. Does the type of adversarial samples (as opposed to less or more challenging/natural cases) influence the conclusion?*
>
> We thank the reviewer for bringing up this point. We realize that the word "adversarial" is not well-suited in Figure (1) and may cause confusion. In our experiments, we do not create synthetic data to deteriorate the generalization performance, but only work with the original data. We change the probability of observing the actual test samples to improve or hurt generalization performance. We will change our wording from "adversarial" to "detrimental" to prevent confusion and provide extra clarification.
>
> However, this comment motivated us to perform another experiment for our final paper. We aim to generate adversarial samples based on the estimator achieved by regression over training samples and use those examples in the test set to perform the same experiment. We expect to see that the gradient ascent increases the weights of those samples which hurts generalization. We thank the reviewer for this idea.
>
>
> - *What are the limitations (computation-wise) of this approach? How well do you think it's adaptable for large (real world) datasets. In case of machine translation for example, the training data can be a couple of million samples.*
>
> We thank the reviewer for bringing this point up, which we should have clearly discussed. Our approach requires solving for the eigenvalues and eigenvectors of the kernel matrices, limiting its applicability to large datasets. However, there are proposed methods to solve kernel regression problem via SGD (e.g. "Scalable Kernel Methods via Doubly Stochastic Gradients" by Dai. et. al. 2015) which can, to some extend, help scaling kernel regression to big data. We will discuss these limitations more in depth.

---

> > ### Comment · Reviewer_U6wb · 2021-09-10
> > **Questions clarified**
> >
> > The rebuttal to me, as well as other reviewers, clarified most of my questions.

---

### Decision · Program_Chairs · 2021-09-27

**Decision:**

Accept (Poster)

**Comment:**

This paper gives a detailed study for out-of-distribution generalization in kernel regression, going substantially beyond known results in this particular problem, which are both of independent interest and also may help serve as a baseline of understanding for similar results in other methods. Some concerns did come up in the reviews both in terms of the framing – which seem to be generally resolved in the discussion phase, though please make sure that these are reflected in the final version of the paper – and in particular for the style of result in Proposition 1, which is worth thinking a little further about. Overall, though, this will be a nice contribution to the conference.